# DATA-EFFICIENT TRAINING BY EVOLVED SAMPLING

## ABSTRACT

Data selection is designed to accelerate learning with preserved performance. To achieve this, a fundamental thought is to identify informative data samples with significant contributions to the training. In this work, we propose **Evolved Sampling** (**ES**), a simple yet effective framework for *dynamic* sampling performed along the training process. This method conducts *batch* level data selection based on *differences* of historical and current losses, significantly reducing the back propagation time with modest additional overheads while maintaining the model performance. Due to its conciseness, ES is readily extensible to incorporate *set* level data selection for further training accelerations. As a plug-and-play framework, ES consistently achieves lossless training accelerations across various models (ResNet, ViT, ALBERT), datasets (CIFAR, ImageNet, GLUE), and optimizers (SGD, Adam), saving up to 40% wall-clock time. Particularly, the improvement is more significant under the *noisy supervision* setting. When there are severe corruptions in labels, ES can obtain accuracy improvements of approximately 20% relative to the standard batched sampling. Our results motivate further investigations on the data efficiency aspect of modern large-scale machine learning.

## 1 INTRODUCTION

Deep learning has showcased remarkable performance across a variety of real-world applications, particularly leading to unparalleled successes of large "foundation" models (Touvron et al., 2023; Rombach et al., 2022). On the other hand, since these large models are usually trained on web-scale datasets, the overall computation and memory loads are considerably increasing, calling for more *efficient* developments of modern machine learning. Efficient learning involves several aspects, centering around models, data, optimization, systems, and so on (Shen et al., 2023).

For data-efficient machine learning, the core is to properly evaluate the importance per data sample in the original (large-scale) datasets. A broad array of methods is applied in a *static* manner, where the samples' importance is determined before the training. By leveraging the feature representations of data (Swayamdipta et al., 2020; Xie et al., 2023b), this importance can be evaluated based on a variety of metrics such as distances (Har-Peled & Mazumdar, 2004; Huang et al., 2023; Bachem et al., 2015; Xia et al., 2023; Abbas et al., 2023; Sorscher et al., 2022), uncertainties (Coleman et al., 2020; Ducoffe & Precioso, 2018; Margatina et al., 2021; Dasgupta et al., 2019; Liu et al., 2021), errors (Toneva et al., 2019; Paul et al., 2021; Langberg & Schulman, 2010; Munteanu et al., 2018), etc, and learned via procedures from the meta optimization (Killamsetty et al., 2021c;b; Jain et al., 2024; Wang et al., 2022) and dataset distillation (Nguyen et al., 2021; Wang et al., 2022; Zhao & Bilen, 2023). However, these approaches can be prohibitively expensive to apply in practice, since their dependence on feature representations requires additional (pre-)training in advance.

Another array of methods lies in a *dynamic* sense, where the samples' importance is simultaneously evaluated along the training process. Dynamic sampling methods can be further divided into two categories: *set* level selection, to prune the whole dataset at the beginning of each epoch (Qin et al., 2024; Raju et al., 2021; Thao Nguyen et al., 2023; Attendu & Corbeil, 2023), and *batch* level selection, to sample subsets from original batches for back propagation (Kawaguchi & Lu, 2020; Katharopoulos & Fleuret, 2017; 2018; Mindermann et al., 2022). Nevertheless, these dynamic sampling methods leverage similar strategies to evaluate the samples' importance. Based on the naive intuition that samples' contributions to the learning are directly associated with gradient updates, it is natural to re-weight data samples with scales of gradients or losses during training. Sampling methods based on the gradients (Mirzasoleiman et al., 2020; Killamsetty et al., 2021a; Hanchi et al., 2022;

Katharopoulos & Fleuret, 2018; Wang et al., 2024) usually suffers from significant computation and memory loads. Sampling methods based on the loss dynamics can involve current losses (Jiang et al., 2019; Loshchilov & Hutter, 2016; Schaul et al., 2016; Kawaguchi & Lu, 2020; Qin et al., 2024; Thao Nguyen et al., 2023; Kumar et al., 2023; Balaban et al., 2023) and historical losses (Attendu & Corbeil, 2023; Raju et al., 2021), and also adopt reference models (Mindermann et al., 2022; Deng et al., 2023; Xie et al., 2023a). However, these approaches exploit the information of losses inadequately by only involving absolute loss values without their "evolutions".

Table 1: The comparison of different dynamic sampling methods. The "history" column denotes whether the method uses historical information along the training. The "robust" column represents the performance robustness under (severe) label noises. The last column summarizes the ratio of samples used for back propagations (BPs) relative to the standard training. Here, $r$ stands for the pruning ratio for *set* level methods (pruning data samples of the whole epoch), and $b/B$ represents the pruning ratio for *batch* level methods (selecting a mini-batch $\mathfrak{b}$ (subset) from a meta-batch $\mathcal{B}$).

| | *set* | *batch* | history | robust | # of samples for BP |
|---|---|---|---|---|---|
| UCB (Raju et al., 2021) | ✓ | | ✓ | | $1 - r$ |
| KA (Thao Nguyen et al., 2023) | ✓ | | | | $1 - r$ |
| InfoBatch (Qin et al., 2024) | ✓ | | ✓ | | $1 - r$ |
| Loss (Katharopoulos & Fleuret, 2017) | | ✓ | | | $b/B$ |
| Order (Kawaguchi & Lu, 2020) | | ✓ | | | $b/B$ |
| ES (ours) | | ✓ | ✓ | ✓ | $b/B$ |
| ESWP (ours) | ✓ | ✓ | ✓ | ✓ | $(1 - r)b/B$ |

To tackle these challeges, we propose a novel dynamic sampling framework, **Evolved Sampling** (**ES**), which incorporates the loss *evolution* or *differences* along the training process to determine samples' importance and conduct *batch* level selection, without the demand of pre-trained reference models. Importantly, ES employs the technique of *decoupled* exponential moving averages (EMAs) to compute sampling weights/probabilities, where one iterative scheme is designed to mix current losses with tracked weights updated by the single EMA over historical losses. Due to its simplicity, this procedure is effortless to implement and only introduces mild computational overheads with negligible memory costs, while significantly reducing the number of samples used for back propagations (BPs) and consequently saving the overall wall-clock time, without degrading the model performance. Moreover, ES facilitates convenient extensions to data pruning on the *set* level, i.e. **Evolved Sampling with Pruning** (**ESWP**), leading to further accelerations with lossless model performance. We demonstrate the differences in details between our proposed methods (ES/ESWP) and previous dynamic sampling methods in Table 1.

Our contributions can be summarized as follows:

- On the theoretical side, we provide justifications that decoupled EMAs applied in ES(WP) introduce additional *differences* of losses across time, which is a certain type of first-order information, and hence appropriately alleviates the effect of loss oscillations on sampling with more sufficient exploitation of the loss dynamics. From another perspective, ES with gradient decent can be viewed as the solution to a distributionally robust optimization (DRO) problem, where the reference objective is approximated via historical losses.

- We carry out extensive experiments to verify the effectiveness, efficiency, robustness and flexibility of ES(WP). It is shown that ES(WP) consistently achieves lossless training accelerations across various models (ResNet, ViT, ALBERT), datasets (CIFAR, ImageNet, GLUE), and optimizers (SGD, Adam), saving up to 40% wall-clock time. In addition, certain hyper-parameters of ES are flexible to be tuned to trade-off between model performance and training costs. Moreover, ES(WP) also exhibits boosted performance particularly under the *noisy supervision* setting: Given severe noises in labels, ES(WP) can obtain significant accuracy improvements of approximately 20% relative to the standard batched sampling.

The rest of this paper is organized as follows. In Section 2, we discuss the related work on static and dynamic sampling. In Section 3, we present the proposed methods, including a comparison with former sampling approaches and corresponding theoretical justifications. Numerical experiments and ablation studies are provided in Section 4. The discussions and outlook are provided in Section 5. All the details of proofs and experiments are found in the appendices.

**Notations.**    For consistency, we adhere to the following notations. Throughout this paper, we use normal letters to denote scalars. Boldfaced lower-case letters are reserved for vectors. We denote the cardinality of a set $S$ by $|S|$. Let $[n] := \{1, 2, \ldots, n\}$ for $n \in \mathbb{N}_+$. Let $\mathbf{1}_n \in \mathbb{R}^n$ be the vector of all ones. For any $c > 0$, $\lceil c \rceil$ represents the smallest positive integer such that $\lceil c \rceil \geq c$. We use the big-O notation $f(t) = \mathcal{O}(g(t))$ to represent that $f$ is bounded above by $g$ asymptotically, i.e., there exists $c > 0, t_0 > 0$ such that $f(t) \leq cg(t)$ for any $t \geq t_0$.

## 2    RELATED WORK

**Static sampling.**    Methods to sampling statically can be based on geometry, uncertainty, error, meta optimization, dataset distillation, etc. With numerous studies on theoretical guarantees (Har-Peled & Mazumdar, 2004; Huang et al., 2023; Bachem et al., 2015), the coreset selection is designed to approximate original datasets with smaller (re-weighted) subsets, typically achieved by clustering in representation spaces (Xia et al., 2023; Abbas et al., 2023; Sorscher et al., 2022). Uncertainty-based methods use probability metrics such as the confidence, entropy (Coleman et al., 2020) and distances to decision boundaries (Ducoffe & Precioso, 2018; Margatina et al., 2021; Dasgupta et al., 2019; Liu et al., 2021). Sampling methods based on errors assume that training samples with more contributions to errors are more important. Errors are evaluated with merics such as forgetting events (Toneva et al., 2019), GRAND & EL2N score (Paul et al., 2021), and sensitivity (Langberg & Schulman, 2010; Munteanu et al., 2018). As is discussed before, these static sampling methods require extra training, leading to considerable costs in both computation and memory.

**Dynamic sampling.**    Methods to sampling dynamically typically leverage metrics based on losses and gradients along the training process. Loss-adaptive sampling re-weights data points during the training according to current losses (Katharopoulos & Fleuret, 2017; Jiang et al., 2019; Loshchilov & Hutter, 2016; Schaul et al., 2016) and historical losses(Oren et al., 2019; Sagawa et al., 2020). To name a few, Ordered SGD (Kawaguchi & Lu, 2020) selects top-$q$ samples in terms of the loss ranking per training step. InfoBatch (Qin et al., 2024) randomly prunes a portion of less informative samples with losses below the average and then re-scales the gradients. KAKURENBO (Thao Nguyen et al., 2023) combines current losses with the prediction accuracy and confidence to design a sampling framework with moving-back. Kumar et al. (2023) and Balaban et al. (2023) assign weights as functions of current losses based on the robust optimization theory. Attendu & Corbeil (2023) and Raju et al. (2021) use the exponential moving average over past losses for sampling. There are also studies adopting reference models, including Mindermann et al. (2022); Deng et al. (2023); Xie et al. (2023a) and so on. These methods either exploit the information of losses inadequately, or require to train additional architectures. Gradient-based sampling methods involve (i) gradient matching, such as CRAIG (Mirzasoleiman et al., 2020) and GRAD-MATCH (Killamsetty et al., 2021a), which approximate the "full" gradients computed on original datasets via the gradients computed on subsets; (ii) gradient adaption, where the sampling probability is basically determined by current scales of gradients (Hanchi et al., 2022; Katharopoulos & Fleuret, 2018). A recent work (Wang et al., 2024) uses a intricate layer-wise sampling scheme with complex variance control. Obviously, gradient-based sampling methods lead to much more computation and memory overheads than loss-based methods.

***Set* level versus *batch* level.**    Dynamic sampling methods can be divided into two categories based on the level where data selection is performed: (i) *set* level selection, to prune the whole dataset at the beginning of each epoch (Qin et al., 2024; Raju et al., 2021; Thao Nguyen et al., 2023; Attendu & Corbeil, 2023); (ii) *batch* level selection, to sample subsets from the original batches for back propagations (Kawaguchi & Lu, 2020; Katharopoulos & Fleuret, 2017; 2018; Mindermann et al., 2022). These two types of methods, facilitating training accelerations from different perspectives, are not mutually exclusive. However, to the best of our knowledge, we are not aware of any algorithms combining both of them.

## 3    METHODS

### 3.1    PRELIMINARIES

The classic setting of general machine learning tasks is as follows. Given a dataset $\mathcal{D} := \{(\boldsymbol{x}_i, y_i)\}_{i=1}^n$ (labeled) or $\mathcal{D} := \{\boldsymbol{x}_i\}_{i=1}^n$ (unlabeled) of size $n \in \mathbb{N}_+$, the goal is to solve the empirical risk

minimization (ERM) problem:

$$\min_{\boldsymbol{\theta} \in \boldsymbol{\Theta}} \hat{L}_n(\boldsymbol{\theta}) := \frac{1}{n} \sum_{i=1}^{n} \ell_i(\boldsymbol{\theta}), \tag{3.1}$$

$$\text{where } \ell_i(\boldsymbol{\theta}) := \ell(f(\boldsymbol{x}_i; \boldsymbol{\theta}), y_i), \quad \text{or } \ell_i(\boldsymbol{\theta}) := \ell(f(\boldsymbol{x}_i; \boldsymbol{\theta})). \tag{3.2}$$

Here, $\ell(\cdot, \cdot)$ or $\ell(\cdot)$ denotes the *non-negative* loss function, and $\hat{L}_n(\boldsymbol{\theta})$ represents the empirical averaged loss over $n$ data samples. When $n$ is large, a common routine is to compute stochastic gradient on a random batch instead of the whole training set. For instance, starting from an initialization $\boldsymbol{\theta}(0) = \boldsymbol{\theta}_0$, the SGD optimizer updates model by

$$\boldsymbol{\theta}(t+1) = \boldsymbol{\theta}(t) - \frac{\eta_t}{B} \sum_{j=1}^{B} \nabla_{\boldsymbol{\theta}} \ell_{i_j}(\boldsymbol{\theta}(t)) \approx \boldsymbol{\theta}(t) - \eta_t \nabla_{\boldsymbol{\theta}} \hat{L}_n(\boldsymbol{\theta}(t)), \tag{3.3}$$

where $\{\eta_t\}_{t \in \mathbb{N}}$ denotes learning rates, $B \in \mathbb{N}_+$ with $B \le n$ is the batch size. The standard sampling method is to draw the batch $\{\boldsymbol{z}_{i_j}\}_{j=1}^{B} \subset \mathcal{D}$ uniformly without replacement for $\lceil n/B \rceil$ iterations in one epoch, which we refer as the standard batched sampling (baseline).

## 3.2 THEORETICAL MOTIVATIONS

Obviously, the standard batched sampling takes equal treatment to data samples. This can be *inefficient* since different samples may have varied importance to the learning task at different stages of training: As the training proceeds, there are inevitably samples that are fitted more accurately compared with the others, leading to lower priority to learn these better-fitted samples in the sequel. Hence, it is necessary to assign *adaptive* weights for data samples during training.

**Convergence acceleration by loss re-weighting.** As is discussed before, it is normal in practice to measure the data samples' importance with scales of losses along the training, which allocates more weights on samples with larger losses. The experiments in Katharopoulos & Fleuret (2017) and Kawaguchi & Lu (2020) have suggested that this kind of "loss-weighted" gradient decent dynamics appears faster convergence in terms of both training and test errors compared to (3.3). To step further, the first contribution of this work is to theoretically develop these former literatures, by proving the following convergence rate.

**Proposition 1** (Reduced version; see a full version in Proposition A.1)**.** *Consider the continuous-time and full-batch idealization of the loss-weighted gradient decent, i.e.*

$$\frac{d}{ds} \hat{\boldsymbol{\theta}}_n^{lw}(s) = -\sum_{i=1}^{n} \frac{\ell_i(\hat{\boldsymbol{\theta}}_n^{lw}(s))}{\sum_{j=1}^{n} \ell_j(\hat{\boldsymbol{\theta}}_n^{lw}(s))} \nabla_{\boldsymbol{\theta}} \ell_i(\hat{\boldsymbol{\theta}}_n^{lw}(s)), \quad \hat{\boldsymbol{\theta}}_n^{lw}(0) = \boldsymbol{\theta}_0. \tag{3.4}$$

*Assume that there exists $\boldsymbol{\theta}^* \in \boldsymbol{\Theta}$ such that $\hat{L}_n(\boldsymbol{\theta}^*) = 0$ and $\ell_i(\cdot)$ is convex for each $i \in [n]$. Then, we have the more-than sub-linear convergence rate of (3.4):*

$$\frac{1}{s} \int_0^s \hat{L}_n(\hat{\boldsymbol{\theta}}_n^{lw}(s'))ds' - \hat{L}_n(\boldsymbol{\theta}^*) \le \frac{1}{2s} \|\boldsymbol{\theta}_0 - \boldsymbol{\theta}^*\|_2^2 - \frac{1}{s} \int_0^s \Delta(s')ds', \quad s > 0, \tag{3.5}$$

*where $\Delta(\cdot)$ is a positive-valued function on $[0, \infty)$.*

Proposition 1 suggests that (under certain regularity conditions) the time-averaged loss of loss-weighted gradient flow converges more than sub-linearly to the global minimum, while the standard gradient flow only has the sub-linear convergence. This theoretical characterization fundamentally gives chances to learning acceleration by leveraging losses in the gradient-based training dynamics.

In general, for any $i \in [n]$ and $t \in \mathbb{N}$, define $w_i(t)$ as the (unnormalized) weight of the $i$-th sample at the $t$-th (training) step. For the standard batched sampling, we obviously have the uniform weights: $w_i(t) \equiv 1/n$. For the loss-weighted sampling corresponding to (3.4), one calculates the sampling probability as

$$p_i(t) \propto w_i(t) = \ell_i(\boldsymbol{\theta}(t)), \tag{3.6}$$

i.e., the weight is set as the current loss value. On top of that, there are also some variants of loss re-weighted sampling strategies: For instance, Kumar et al. (2023) sets $w_i(t) = g(\ell_i(\boldsymbol{\theta}(t)))$, where the function $g(\cdot)$ is pre-defined based on the theory of robust optimization; Kawaguchi & Lu (2020) directly selects top-$q$ samples in terms of current losses per training step, which can be regarded as another realization of Kumar et al. (2023).

### 3.3 EVOLVED SAMPLING

In general machine learning tasks, the typical behaviors of loss curves often appear decent trends overall, but can oscillate meanwhile due to certain noises. This introduces the sensitivity or instability issue of the sampling scheme (3.6). A commonly-used smoothing operation is the exponential moving average (EMA) of losses

$$p_i(t) \propto w_i(t) = \beta w_i(t-1) + (1-\beta)\ell_i(\boldsymbol{\theta}(t)), \quad w_i(0) = 1/n \tag{3.7}$$

for any $i \in [n]$ and $t \in \mathbb{N}$, where the hyper-parameter $\beta \in [0, 1]$ is typically selected close to 1 to capture more historical information.[1] However, the EMA can potentially erase too many dynamical details (including noises) shown in the loss dynamics. To see this, we give an illustration in Figure 1. The black curve denotes a (polynomially) decayed function with random perturbations, which is designed to mimic typical behaviors of loss curves in general machine learning tasks and fails to provide information robustly due the noises. On the other hand, the blue curve represents the EMA, which leads to over-smoothing due to the average effect.

**Decoupled EMA.** To sufficiently leverage the loss dynamics in a more robust sense, we propose to calculate the sampling probability as

$$\begin{aligned} p_i(t) \propto w_i(t) &= \beta_1 s_i(t-1) + (1-\beta_1)\ell_i(\boldsymbol{\theta}(t)), \\ s_i(t) &= \beta_2 s_i(t-1) + (1-\beta_2)\ell_i(\boldsymbol{\theta}(t)), \quad s_i(0) = 1/n \end{aligned} \tag{3.8}$$

with $\beta_1, \beta_2 \in [0, 1]$ as two hyper-parameters. Here, the intermediate series $\{s_i(t)\}_{t \in \mathbb{N}}$, updated in the EMA scheme, is also referred as the score (for the $i$-th sample). The scheme (3.8) is the so-called *decoupled EMA*,[2] which reduces to (3.7) when $\beta_1 = \beta_2 = \beta$. In Figure 1, it is shown by the red curve and appears an "interpolation" between the original loss and single EMA: When losses oscillate, the decoupled EMA reacts moderately by not only capturing detailed dynamics of losses, but also remaining necessary robustness, exhibiting the flexibility to trade-off (by tuning two betas).

Intuitively, by setting $(\beta_1, \beta_2) \to (0^+, 1^-)$, we are able to exploit the long-term historical information along the training (via $\beta_2$), while focusing on the importance of current losses (via $\beta_1$) and thus can get the best of both world. This simple and elegant design turns out to be surprisingly beneficial in practice, which is further verified in numerous experiments in Section 4.

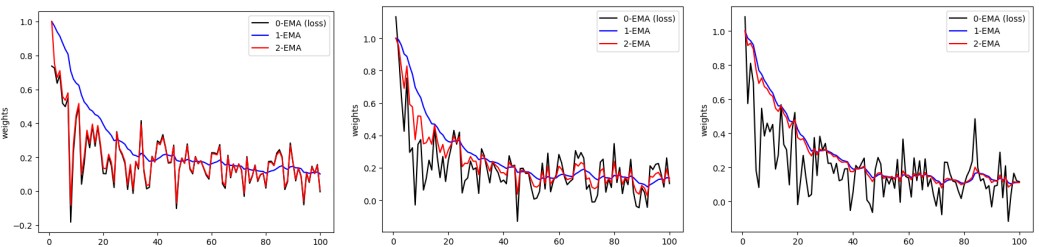

Figure 1: The effect of EMAs, where the output weight is a function of the time step $t$. From left to right: $\beta_1 = 0.1, 0.5, 0.8$, and $\beta = \beta_2 \equiv 0.9$.

**Annealing.** Notably, similar to other loss-weighted sampling methods, the decoupled EMA sampling scheme (3.8) also assigns different weights on the respective gradient of data samples, leading to a biased estimation on the true gradient $\nabla_{\boldsymbol{\theta}} \hat{L}_n(\cdot)$ (that assigns uniform weights). Inspired by Qin et al. (2024), we adopt the *annealing* strategy, to perform normal training (with the standard batched sampling, no data selection) at the last few epochs. Besides, to get a better initialization of the score $\{s_i(\cdot)\}_{i \in [n]}$, we also apply the annealing strategy at the first few epochs.

Combining the decoupled EMA sampling scheme (3.8) with the annealing strategy, we obtain the **Evolved Sampling** (**ES**) framework (formalized in Algorithm 1).

**Pruning.** Note that applying the decoupled EMA sampling scheme (3.8) to meta-batches (with the batch size $B$) has already introduced data selection in a *batch* level, since one can always select a

---

[1]The EMA can be viewed as a weighted average over past $1/(1-\beta)$ moments (when $\beta \to 1^-$).

[2]As a comparison, (3.7) is also called the single EMA. Note that (3.7) reduces to (3.6) when setting $\beta = 0$, and (3.7) reduces to the standard batched sampling when setting $\beta = 1$.

smaller batch (with the batch size $b < B$) out of the meta-batch, according to the sampling probability $p_i(t)$ defined in (3.8). For more aggressive data pruning and enhanced data efficiency, we can further extend ES by involving the *set* level data selection (i.e. randomly pruning the whole dataset according to the probability proportional to the score $\{s_i(e)\}_{i=1}^n$ at the beginning of the $e$-th epoch), which is **Evolved Sampling with Pruning** (**ESWP**; formalized in Algorithm 1).

---

**Algorithm 1** Learning by Evolved Sampling (with Pruning)

---

**Require:** Dataset $\mathcal{D} = \{z_i\}_{i=1}^n$, model space $\boldsymbol{\Theta} \ni \boldsymbol{\theta}$, optimizer (e.g. SGD, Adam)
**Require:** Pruning ratio $r$, meta-batch size $B$, mini-batch size $b \leq B$, decoupled EMAs' hyper-parameters $\beta_1, \beta_2 \in (0, 1)$, total number of epochs $E$, number of annealing epochs $E_a$
  Initialize the model $\boldsymbol{\theta}(0) = \boldsymbol{\theta}_0$, the score $\boldsymbol{s}(0) = \frac{1}{|\mathcal{D}|}\mathbf{1}_n = \frac{1}{n}\mathbf{1}_n$, $t = 0$
  **for** $e = 0, 1, \cdots, E - 1$ **do**
    **if** $E_a \leq e < E - E_a$ **then**
      Sample a sub-dataset $\mathcal{D}_e$ ($|\mathcal{D}_e| = (1 - r)|\mathcal{D}|$) from $\mathcal{D}$ without replacement, according to the
      probability $p_i'(e) \propto s_i(e)$ (normalized w.r.t. $i \in [n]$)                                    $\triangleright$ "pruning"
    **else**
      Set $\mathcal{D}_e = \mathcal{D}$
    **end if**
    **for** $j = 0, 1, \cdots, \lceil \frac{|\mathcal{D}_e|}{B} \rceil - 1$ **do**
      Sample a meta-batch $\mathcal{B}_t$ ($|\mathcal{B}_t| = B$) uniformly from $\mathcal{D}_e$ without replacement
      Compute the loss $\ell_i(\boldsymbol{\theta}(t))$ for $z_i \in \mathcal{B}_t$
      Update the score: $s_i(e + 1) \leftarrow \beta_2 s_i(e) + (1 - \beta_2)\ell_i(\boldsymbol{\theta}(t))$ for $z_i \in \mathcal{B}_t$
      Update the weight: $w_i(e) \leftarrow \beta_1 s_i(e) + (1 - \beta_1)\ell_i(\boldsymbol{\theta}(t))$ for $z_i \in \mathcal{B}_t$
      **if** $E_a \leq e < E - E_a$ **then**
        Sampling a mini-batch $\mathfrak{b}_t$ ($|\mathfrak{b}_t| = b$) from $\mathcal{B}_t$ without replacement, according to the
        probability $p_i(e) \propto w_i(e)$ (normalized w.r.t. $\{i \in \mathbb{N}_+ : z_i \in \mathcal{B}_t\}$)
        Update the model: $\boldsymbol{\theta}(t + 1) \leftarrow \text{optimizer}(\boldsymbol{\theta}(t); \mathfrak{b}_t)$
      **else**
        Update the model: $\boldsymbol{\theta}(t + 1) \leftarrow \text{optimizer}(\boldsymbol{\theta}(t); \mathcal{B}_t)$              $\triangleright$ "annealing"
      **end if**
      $t \leftarrow t + 1$
    **end for**
  **end for**

---

We illustrate the ES(WP) framework (Algorithm 1) in Figure 2. For the essential differences between ES(WP) and previous dynamic sampling methods, one can refer to the taxonomy outlined in Table 1.

**Remark 1.** *Here, we allow the randomness to keep samples with lower weights in the training, which reduces the biases compared to directly discarding them. In addition, there is no need for the probability-based sampling to sort the scores/weights,[3] further reducing the time complexity.*

### 3.4 THEORETICAL JUSTIFICATIONS

We demonstrate theoretically the effectiveness of ES from two aspects. For simplicity, we consider the full-batch case when $B = |\mathcal{D}| = n$, and focus on the core sampling scheme via decoupled EMAs.

**(i) Decoupled EMAs introduce losses'** *differences* **across time.** The first advantage of decoupled EMAs (over single EMAs) can be characterized by the following proposition.

**Proposition 2.** *For any $i \in [n]$, $t \in \mathbb{N}$ and any $\beta_2 \in (0, 1)$, we have*

$$w_i(t) = s_i(t) + \frac{\beta_2 - \beta_1}{1 - \beta_2}(s_i(t) - s_i(t - 1)) \tag{3.9}$$

$$= (1 - \beta_2)\sum_{k=1}^{t} \beta_2^{t-k}\ell_i(\boldsymbol{\theta}(k)) + (\beta_2 - \beta_1)\sum_{k=1}^{t-1} \beta_2^{t-1-k}(\ell_i(\boldsymbol{\theta}(k + 1)) - \ell_i(\boldsymbol{\theta}(k))) + \mathcal{O}(\beta_2^t). \tag{3.10}$$

---

[3]One of the main weaknesses of sorting is its sensitivity to noises, since noisy samples always possess larger losses and can be selected with higher probabilities. As is shown in Section 4.2, Ordered SGD selects many samples with noisy labels as the training proceeds, leading to sub-optimal performance.

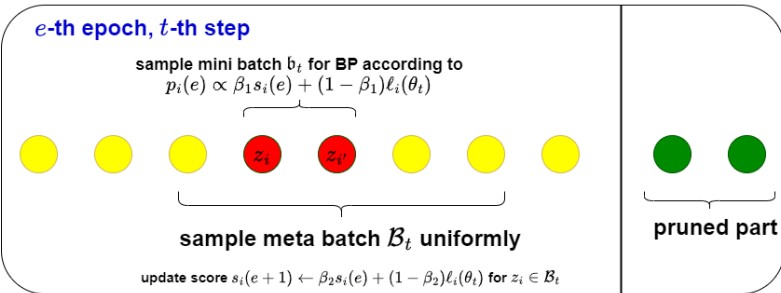

Figure 2: An illustration of ES(WP). At the beginning of the $e$-th epoch, we randomly prune the whole dataset according to the probability proportional to the score $\{s_i(e)\}_{i=1}^n$. At the $t$-th step, we first sample a meta-batch $\mathcal{B}_t$ uniformly without replacement from the remaining dataset, from which we then sample a mini-batch $\mathfrak{b}_t$ for BP, according to the probability defined by decoupled EMA. Note that the scores of samples are updated using the *latest* model parameters. At the first/last few epochs, we use the annealing strategy, i.e. the standard batched sampling without data selection.

The proof of Proposition 2 is deferred to Appendix A.2. Proposition 2 can be understood as follows. The equality (3.9) implies that the improvement of decoupled EMAs (i.e. $\boldsymbol{w}(t)$) over single EMAs (i.e. $\boldsymbol{s}(t)$) is measured by *differences* of single EMAs across time. Since the single EMA leads to in fact a convolution between hyper-parameters' powers and historical losses, the differences across time are also summarized in the convolution between hyper-parameters and losses. Specifically, let $(D\boldsymbol{l})(t) := \boldsymbol{l}(t+1) - \boldsymbol{l}(t)$, $t \in \mathbb{N}$ denote the loss difference across time, then (3.10) can be written as

$$\boldsymbol{w} \approx (1 - \beta_2)(\boldsymbol{\beta}_2 * \boldsymbol{l}) + (\beta_2 - \beta_1)(\boldsymbol{\beta}_2 * (D\boldsymbol{l})), \tag{3.11}$$

where $\boldsymbol{w} := [\boldsymbol{w}(t)]_{t \in \mathbb{N}}$, $\boldsymbol{l} := [\boldsymbol{l}(t)]_{t \in \mathbb{N}}$, $D\boldsymbol{l} := [(D\boldsymbol{l})(t)]_{t \in \mathbb{N}}$ and $\boldsymbol{\beta}_2 := [\beta_2^t]_{t \in \mathbb{N}}$ (boldfaced notations collecting corresponding indexes (i.e. $i \in [n]$)). The common convolution $*$ is operated across time.

**Remark 2.** *When setting $\beta_1 = \beta_2$, the decoupled EMA is reduced to a single EMA, where the first order information of losses (i.e. the second term of RHS of (3.11)) vanishes.*

**(ii) ES to solve a DRO problem.** From another perspective, ES can be also reformulated as a solution to the minimax problem

$$\min_{\boldsymbol{\theta} \in \Theta} \max_{\boldsymbol{p} \in \Delta^n} L_n(\boldsymbol{\theta}; \boldsymbol{p}) := \sum_{i=1}^n p_i(\ell_i(\boldsymbol{\theta}) - \ell_i^{\text{ref}}), \tag{3.12}$$

where $\Delta^n$ denotes the $(n-1)$-dimensional probability simplex. This objective leads to a stronger requirement for robust performances on both typical and rare samples compared to the regular ERM (Shalev-Shwartz & Wexler, 2016). Different from traditional DRO, (3.12) introduces a reference loss $\ell_i^{\text{ref}}$, with the excess loss $\ell_i(\boldsymbol{\theta}) - \ell_i^{\text{ref}}$ measuring the improvement of the model on the $i$-th sample with respect to a reference model (typically *pre-trained*; see e.g. Oren et al. (2019); Xie et al. (2023a); Mindermann et al. (2022)). The second advantage of ES is to naturally leverage losses of historical models along the training dynamics as a proxy of the reference loss $\ell_i^{\text{ref}}$ in (3.12), which can be continuously updated without explicitly (pre-)training additional models.

Specifically, we have the following proposition, and its proof is deferred to Appendix A.3.

**Proposition 3.** *Consider to solve the minimax objective (3.12) via gradient ascent-descent*

$$\begin{cases} \boldsymbol{p}(t) \propto \boldsymbol{w}(t) := \boldsymbol{w}(t-1) + (1-\beta_1)(\boldsymbol{\ell}(\boldsymbol{\theta}(t)) - \boldsymbol{\ell}^{ref}(\boldsymbol{\theta}(1:t-1))), \\ \boldsymbol{\theta}(t+1) := \boldsymbol{\theta}(t) - \eta_t^{\boldsymbol{\theta}} \sum_{i=1}^n p_i(t) \nabla_{\boldsymbol{\theta}} \ell_i(\boldsymbol{\theta}(t)), \end{cases} \tag{3.13}$$

*where the reference loss is defined as $\boldsymbol{\ell}^{ref}(\boldsymbol{\theta}(1:t)) := [\ell_i^{ref}(\boldsymbol{\theta}(1:t))]_{i \in [n]}$ with $\ell_i^{ref}(\boldsymbol{\theta}(1:t)) := \frac{1-2\beta_1+\beta_1\beta_2}{1-\beta_1}\ell_i(\boldsymbol{\theta}(t)) + \frac{\beta_1(1-\beta_2)^2}{1-\beta_1}\sum_{k=1}^{t-1}\beta_2^{t-1-k}\ell_i(\boldsymbol{\theta}(k)) + \frac{\beta_1(1-\beta_2)\beta_2^{t-1}}{n(1-\beta_1)}$, $i \in [n]$. Then, the dynamics (3.13) is consistent with gradient descent sampled with the decoupled EMA (3.8).*

# 4 EXPERIMENTS

In this section, we provide numerical simulations on the proposed method (ES(WP); Algorithm 1) to demonstrate its effectiveness, efficiency, robustness and flexibility.

## 4.1 EFFECTIVENESS AND EFFICIENCY

We compare the proposed methods ES/ESWP, with a group of former dynamic sampling approaches, including the standard batched sampling (Baseline), Order (Kawaguchi & Lu, 2020)), Loss (Katharopoulos & Fleuret, 2017), InfoBatch (Qin et al., 2024), KA (Thao Nguyen et al., 2023), UCB (Raju et al., 2021). For all sampling methods, the hyper-parameters used in data augmentation, tokenization are maintained the same (see more details in Appendix B). All the reported results are evaluated on the average of 2-4 independent random trials.

**Configurations.** For ES/ESWP, the default hyper-parameters are as follows: The annealing ratio is $E_a/E = 5\%$; the pruning ratio is $r = 20\%$ for ESWP; in decoupled EMAs, $(\beta_1, \beta_2) = (0.2, 0.9)$ for ES, $(\beta_1, \beta_2) = (0.2, 0.8)$ for ESWP; for both ES and ESWP, the ratio of mini-batch size over meta-batch size is $b/B = 25\%$. For the two *batch* level selection methods (Order, Loss), we use the same mini/meta-batch size. For InfoBatch, KA and UCB, we use the default hyper-parameters in original papers. For computer vision (CV) tasks, we train ResNet-18/50 (R-18/50) models on CIFAR-10/100, using SGD for 200 epochs, where the meta-batch size $B = 128/256$ for ResNet-18/50 ($b/B = 50\%$ for ResNet-50). We also fine-tune the ViT-Large (Dosovitskiy et al., 2021) on the ImageNet-1K dataset, using Adam for 10 epochs, where the meta-batch size $B = 256$. For natural language processing (NLP) tasks, we fine-tune the ALBERT-Base-v2 (Lan et al., 2020) model on the GLUE benchmark, using AdamW for 10 epochs, where $B$ is set according to Xie et al. (2023b).

**Results.** We report the test classification accuracy and overall wall-clock time for the evaluation of both effectiveness and efficiency. The results are as follows.

(i) For small-scale tasks, we train ResNet models on CIFAR datasets, and summarize the performance of different sampling methods in Table 2. It is shown that the batch level selection methods (Loss, Order, ES) typically exhibits limited accelerations on these small-scale tasks, since these methods often require additional forward propagation overheads that are not negligible compared to BPs. Nevertheless, ES is the only algorithm that achieves lossless accelerations across all methods. Notably, ESWP saves the most computation time while maintaining the best performance (also comparable to Baseline) among set level selection methods (UCB, KA, InfoBatch).

Table 2: The test accuracy (%) and saved time of training ResNet models on CIFAR datasets.

| | CIFAR-10 (R-18) | | CIFAR-100 (R-18) | | CIFAR-100 (R-50) | |
|---|---|---|---|---|---|---|
| Baseline | 95.4 | | 78.8 | | 81.1 | |
| UCB (Raju et al., 2021) | $95.2_{\downarrow 0.2}$ | 18% | $77.6_{\downarrow 1.2}$ | 18% | $80.5_{\downarrow 0.6}$ | 24% |
| KA (Thao Nguyen et al., 2023) | $95.3_{\downarrow 0.1}$ | 21% | $78.1_{\downarrow 0.7}$ | 21% | $80.2_{\downarrow 0.9}$ | 24% |
| InfoBatch (Qin et al., 2024) | $95.3_{\downarrow 0.1}$ | 21% | $78.4_{\downarrow 0.4}$ | **24%** | $80.4_{\downarrow 0.7}$ | 28% |
| Loss (Katharopoulos & Fleuret, 2017) | $95.3_{\downarrow 0.1}$ | 11% | $78.4_{\downarrow 0.4}$ | 10% | $80.5_{\downarrow 0.6}$ | 12% |
| Order (Kawaguchi & Lu, 2020) | $\mathbf{95.4}_{\uparrow 0.0}$ | 11% | $78.5_{\downarrow 0.3}$ | 10% | $80.9_{\downarrow 0.2}$ | 12% |
| ES | $\mathbf{95.4}_{\uparrow 0.0}$ | 10% | $\mathbf{78.8}_{\uparrow 0.0}$ | 10% | $\mathbf{81.1}_{\uparrow 0.0}$ | 11% |
| ESWP | $95.3_{\downarrow 0.1}$ | **24%** | $78.6_{\downarrow 0.2}$ | **24%** | $80.6_{\downarrow 0.5}$ | **31%** |

(ii) For large-scale tasks, we fine-tune the ViT-Large model on the ImageNet-1K dataset, and summarize the performance of different sampling methods in Table 3. Under this setting, ES continues to show the best performance among batch level selection methods and the second-to-highest accuracy across all sampling methods. Notably, ESWP achieves the best performance and most significant time reduction, suggesting that ESWP inherits the advantages of *both* set and batch level selection methods. In addition, it is observed that the training speed-up of batch level methods gets far more significant given these large-scale tasks, conversely surpassing the set level methods compared to (i). This is due to the dominance of the saved computation in BPs. Furthermore, many sampling methods achieve higher accuracies than the baseline, implying huge potentials of data selection in large-scale machine learning.

Table 3: The validation accuracy (%) and saved time of fine-tuning ViT-Large on the ImageNet-1K.

| | Baseline | UCB | KA | InfoBatch | Loss | Order | ES | ESWP |
|---|---|---|---|---|---|---|---|---|
| Accuracy | 84.4 | 84.2 | 84.3 | 84.7 | 84.3 | 84.2 | 84.7 | **85.0** |
| Time↓ | - | 23.6% | 25.3% | 23.5% | 36.4% | 38.2% | 26.0% | **40.7%** |

(iii) For NLP tasks, we fine-tune the ALBERT-Base model on the GLUE benchmark, and summarize the the performance of different sampling methods in Table 4. On most of the datasets and in the averaged sense, ES/ESWP outperforms all the other sampling methods, and shows improved performance over the baseline with substantial reduction of computation time.

Table 4: The validation metric (%) and saved time of fine-tuning the ALBERT-Base on the GLUE.

|  | CoLA | SST2 | QNLI | QQP | MNLI-m | MRPC | RTE | STSB | Avg. | Time↓ |
|---|---|---|---|---|---|---|---|---|---|---|
| Baseline | 56.7 | 92.2 | 91.1 | **90.3** | **84.7** | 88.5 | 74.0 | 89.6 | 83.4 | - |
| InfoBatch | 57.9 | 92.1 | 91.2 | **90.3** | 84.5 | 89.2 | 73.8 | **89.7** | $83.6_{\uparrow 0.2}$ | 28.3% |
| Loss | 55.1 | 92.3 | 91.4 | 90.2 | 84.4 | 88.6 | 69.6 | 89.5 | $82.6_{\downarrow 0.8}$ | 20.8% |
| Order | 55.4 | 92.6 | 91.3 | 90.1 | 80.9 | 84.6 | 63.2 | 89.4 | $80.9_{\downarrow 2.5}$ | 20.8% |
| ES | **58.4** | 92.4 | 91.4 | 90.2 | 84.5 | 88.7 | **75.8** | 89.6 | $\mathbf{83.9}_{\uparrow 0.5}$ | 20.2% |
| ESWP | 57.5 | **93.1** | **91.7** | 90.0 | **84.7** | **89.8** | 72.8 | 89.4 | $83.6_{\uparrow 0.2}$ | **33.1%** |

## 4.2 ROBUSTNESS UNDER LABEL NOISES

In this section, we further demonstrate that ES(WP) exhibits more notable advantages when there are label noises. We train ResNet models on CIFAR datasets under both light (10%) and heavy (40%) label noises, which are injected randomly with uniform probabilities or flipped to another class (see details in Appendix B.1). The other configurations remain the same as those in Section 4.1, except that we only train 100 epochs to avoid the overfitting of ResNet-50. The results are as follows.

(i) In Table 5, we summarize the results of training the ResNet-18 model on the CIFAR-100 dataset under different levels and types of label noises. It is shown that ES/ESWP consistently outperforms all the other sampling methods (including the baseline) with clear gaps, and the improvement is more significant when the label noises become severer.

Table 5: The test accuracy (%) of training the ResNet-18 on the CIFAR-100 with label noises.

|  | Baseline | UCB | KA | InfoBatch | Loss | Order | ES | ESWP |
|---|---|---|---|---|---|---|---|---|
| Clean (0%) | 78.8 | $77.6_{\downarrow 1.2}$ | $78.1_{\downarrow 0.7}$ | $78.4_{\downarrow 0.4}$ | $78.4_{\downarrow 0.4}$ | $78.5_{\downarrow 0.3}$ | $\mathbf{78.8}_{\uparrow 0.0}$ | $78.6_{\downarrow 0.2}$ |
| Flip (10%) | 72.3 | $68.7_{\downarrow 3.6}$ | $67.0_{\downarrow 5.3}$ | $71.5_{\downarrow 0.8}$ | $72.9_{\uparrow 0.6}$ | $70.8_{\downarrow 1.5}$ | $73.1_{\uparrow 0.8}$ | $\mathbf{73.1}_{\uparrow 0.8}$ |
| Flip (40%) | 46.8 | $43.9_{\downarrow 2.9}$ | $45.0_{\downarrow 1.8}$ | $46.6_{\downarrow 0.2}$ | $53.6_{\uparrow 6.8}$ | $47.8_{\uparrow 1.0}$ | $57.1_{\uparrow 10.3}$ | $\mathbf{58.2}_{\uparrow 11.4}$ |
| Uniform (10%) | 68.3 | $66.6_{\downarrow 1.7}$ | $65.4_{\downarrow 2.9}$ | $67.8_{\downarrow 0.5}$ | $67.0_{\downarrow 1.3}$ | $65.4_{\downarrow 2.9}$ | $\mathbf{68.7}_{\uparrow 0.4}$ | $68.7_{\uparrow 0.4}$ |
| Uniform (40%) | 50.8 | $44.1_{\downarrow 6.7}$ | $44.0_{\downarrow 6.8}$ | $50.8_{\uparrow 0.0}$ | $57.3_{\uparrow 6.5}$ | $37.9_{\downarrow 12.9}$ | $\mathbf{61.1}_{\uparrow 10.3}$ | $60.1_{\uparrow 9.3}$ |

(ii) For further verifications and analysis, we also plot the test accuracy versus wall-clock time in Figure 3(a) when training the ResNet-50 model on the CIFAR-100 dataset with 40% uniform label noises. It is observed that ES/ESWP achieves the significantly improved performance and considerable accelerations. To further understand the underlying mechanism why ES/ESWP works, we plot the relative magnitudes of gradients evaluated on corrupted samples over the whole mini-batch at each training iteration in Figure 3(b). It turns out that ES/ESWP exhibits a smaller portion of noisy gradients along the training (particularly at the convergence stage), suggesting that ES/ESWP selects proper samples for training given severe label corruptions.

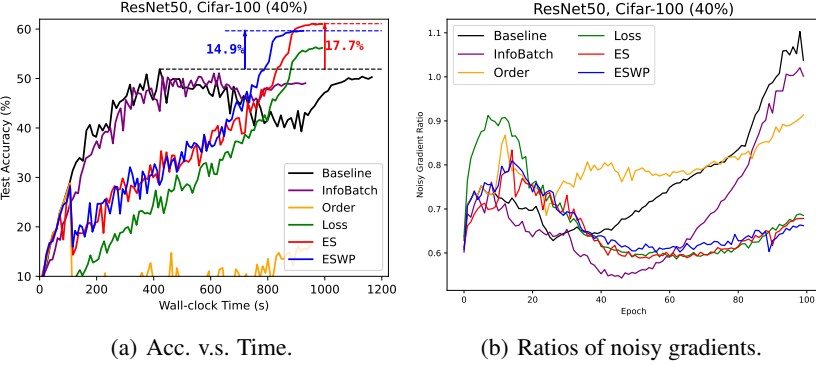

(a) Acc. v.s. Time.  (b) Ratios of noisy gradients.

Figure 3: Results of training the ResNet-50 on the CIFAR-100 with 40% uniform label noises.

### 4.3 ABLATION STUDIES

**Decoupled EMA and annealing.** We numerically test the effectiveness of two important components applied in ES, i.e. the decoupled EMA and annealing. Here, we perform ablations on combinations of "Loss", "A" (Annealing), "E" (single EMA) and "DE" (decoupled EMA). From Table 6, it is observed that: (i) Annealing is an effective technique to boost performance; (ii) EMA also contributes to the improvements; (iii) Compared to the single EMA, the decoupled EMA provides more substantial benefits to the training process.

Table 6: Ablations on decoupled EMAs and annealing for different models, datasets and noises.

|  | ResNet-18 | | ResNet-50 | | ALBERT-Base |
|---|---|---|---|---|---|
|  | CIFAR-10 (40%) | CIFAR-100 (10%) | CIFAR-100 | CIFAR-100 (40%) | CoLA |
| Loss | 83.3 | 67.0 | 80.5 | 53.8 | 55.1 |
| Loss + A | 84.4 | 68.4 | 80.8 | 60.1 | 55.8 |
| Loss + E | 83.4 | 66.2 | 80.5 | 53.6 | 57.6 |
| Loss + DE | 83.7 | 66.8 | **81.1** | 54.2 | 57.5 |
| Loss + A + E | 84.6 | 68.0 | 80.4 | 60.3 | 57.6 |
| ES = Loss + A + DE | **85.2** | **68.7** | **81.1** | **60.9** | **58.4** |

**Trade-offs between accuracies and accelerations.** We emphasize that the ratio (of batch sizes) $b/B$ in ES is user-defined, and is flexible to be tuned to trade-off between model performance and training costs. We evaluate different values of $b/B$ when fine-tuning ViT-Large on the ImageNet-1K, and plot the results in Figure 4. It is shown that ES robustly achieves lossless performance when $b/B \geq 1/16$. When the data selection is too aggressive ($b/B \leq 1/32$), the performance degrades as expected, primarily due to the increase in variances of stochastic gradients.

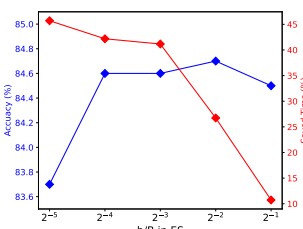

Figure 4: The effect of $b/B$.

**Choices of $(\beta_1, \beta_2)$.** To investigate the impact of newly introduced hyper-parameters in ES, we test different choices of $(\beta_1, \beta_2)$ when training ResNet-18 on CIFAR and ALBERT-Base on CoLA. The results shown in Figure 5 further verify the "optimality" of default configurations to set $(\beta_1, \beta_2) \rightarrow (0^+, 1^-)$, allowing for guidance by current losses while exploiting long-term historical information.

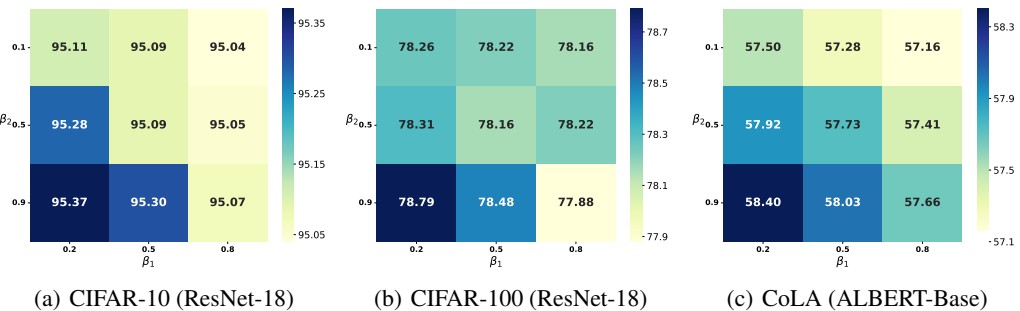

(a) CIFAR-10 (ResNet-18)  (b) CIFAR-100 (ResNet-18)  (c) CoLA (ALBERT-Base)

Figure 5: The effect of $(\beta_1, \beta_2)$ on the performance (test accuracy (%)).

### 5 CONCLUSION

In this work, we propose a simple yet effective framework, Evolved Sampling, which can be applied to general machine learning tasks to improve the data efficiency in a dynamic manner. By further adopting differences of historical losses to determine samples' importance for data selection, Evolved Sampling can achieve lossless training with significant accelerations, particularly when there are severe noises in labels. Studies in the future may include three aspects: (i) More rigorous mathematical analysis on the effect of data selection (e.g. Kolossov et al. (2024)); (ii) More specific applications, such as data selection/reduction on domain mixtures (e.g. Chen et al. (2023); Xie et al. (2023a)); (iii) More efficient and scalable implementation, such as data parallelism (You et al., 2017; 2020). These directions are certainly worthy of explorations in the future.

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

## A PROOFS

### A.1 PROOF OF PROPOSITION 1

**Proposition A.1** (A full version of Proposition 1). *Consider the continuous-time and full-batch idealization of the gradient decent, i.e. the standard gradient flow training dynamics*

$$\frac{d}{dt}\hat{\boldsymbol{\theta}}_n(t) = -\nabla_{\boldsymbol{\theta}}\hat{L}_n(\hat{\boldsymbol{\theta}}_n(t)) = -\frac{1}{n}\sum_{i=1}^n \nabla_{\boldsymbol{\theta}}\ell_i(\hat{\boldsymbol{\theta}}_n(t)), \quad \hat{\boldsymbol{\theta}}_n(0) = \boldsymbol{\theta}_0, \tag{A.1}$$

*and its loss-weighted variant*

$$\frac{d}{ds}\hat{\boldsymbol{\theta}}_n^{lw}(s) = -\sum_{i=1}^n \frac{\ell_i(\hat{\boldsymbol{\theta}}_n^{lw}(s))}{\sum_{j=1}^n \ell_j(\hat{\boldsymbol{\theta}}_n^{lw}(s))}\nabla_{\boldsymbol{\theta}}\ell_i(\hat{\boldsymbol{\theta}}_n^{lw}(s)), \quad \hat{\boldsymbol{\theta}}_n^{lw}(0) = \boldsymbol{\theta}_0. \tag{A.2}$$

*Assume that there exists $\boldsymbol{\theta}^* \in \Theta$ such that $\hat{L}_n(\boldsymbol{\theta}^*) = 0$,[4] and $\ell_i(\cdot)$ is convex for each $i \in [n]$. Then, we have the more-than sub-linear convergence rate of (A.2):*

$$\frac{1}{s}\int_0^s \hat{L}_n(\hat{\boldsymbol{\theta}}_n^{lw}(s'))ds' - \hat{L}_n(\boldsymbol{\theta}^*) \leq \frac{1}{2s}\|\boldsymbol{\theta}_0 - \boldsymbol{\theta}^*\|_2^2 - \frac{1}{s}\int_0^s \Delta(s')ds', \quad s > 0, \tag{A.3}$$

*where $\Delta(\cdot)$ is a positive-valued function on $[0,\infty)$. Moreover, for any $s, t \geq 0$ such that $\hat{L}_n(\hat{\boldsymbol{\theta}}_n(t)) = \hat{L}_n(\hat{\boldsymbol{\theta}}_n^{lw}(s)) \triangleq l \geq 0$,[5] we have*

$$\frac{d}{ds}\|\hat{\boldsymbol{\theta}}_n^{lw}(s) - \boldsymbol{\theta}^*\|_2^2 \leq -2\left(l + \Delta(s)\right), \tag{A.4}$$

$$\frac{d}{dt}\|\hat{\boldsymbol{\theta}}_n(t) - \boldsymbol{\theta}^*\|_2^2 \leq -2l. \tag{A.5}$$

*Proof.* For any $\boldsymbol{\theta} \in \Theta$, we have

$$\frac{d}{dt}\|\hat{\boldsymbol{\theta}}_n(t) - \boldsymbol{\theta}\|_2^2 = 2\left\langle \hat{\boldsymbol{\theta}}_n(t) - \boldsymbol{\theta}, \frac{d}{dt}\hat{\boldsymbol{\theta}}_n(t)\right\rangle$$

$$= \frac{2}{n}\sum_{i=1}^n \left\langle \boldsymbol{\theta} - \hat{\boldsymbol{\theta}}_n(t), \nabla_{\boldsymbol{\theta}}\ell_i(\hat{\boldsymbol{\theta}}_n(t))\right\rangle$$

$$\leq \frac{2}{n}\sum_{i=1}^n \left(\ell_i(\boldsymbol{\theta}) - \ell_i(\hat{\boldsymbol{\theta}}_n(t))\right), \tag{A.6}$$

and

$$\frac{d}{ds}\|\hat{\boldsymbol{\theta}}_n^{lw}(s) - \boldsymbol{\theta}\|_2^2 = 2\left\langle \hat{\boldsymbol{\theta}}_n^{lw}(s) - \boldsymbol{\theta}, \frac{d}{ds}\hat{\boldsymbol{\theta}}_n^{lw}(s)\right\rangle$$

$$= 2\sum_{i=1}^n \frac{\ell_i(\hat{\boldsymbol{\theta}}_n^{lw}(s))}{\sum_{j=1}^n \ell_j(\hat{\boldsymbol{\theta}}_n^{lw}(s))}\left\langle \boldsymbol{\theta} - \hat{\boldsymbol{\theta}}_n^{lw}(s), \nabla_{\boldsymbol{\theta}}\ell_i(\hat{\boldsymbol{\theta}}_n^{lw}(s))\right\rangle$$

$$\leq 2\sum_{i=1}^n \frac{\ell_i(\hat{\boldsymbol{\theta}}_n^{lw}(s))}{\sum_{j=1}^n \ell_j(\hat{\boldsymbol{\theta}}_n^{lw}(s))}\left(\ell_i(\boldsymbol{\theta}) - \ell_i(\hat{\boldsymbol{\theta}}_n^{lw}(s))\right). \tag{A.7}$$

Note that

$$\sum_{i=1}^n \left[\frac{\ell_i(\hat{\boldsymbol{\theta}}_n^{lw}(s))}{\sum_{j=1}^n \ell_j(\hat{\boldsymbol{\theta}}_n^{lw}(s))}\left(\ell_i(\boldsymbol{\theta}) - \ell_i(\hat{\boldsymbol{\theta}}_n^{lw}(s))\right) - \frac{1}{n}\left(\ell_i(\boldsymbol{\theta}) - \ell_i(\hat{\boldsymbol{\theta}}_n(t))\right)\right]$$

$$= \sum_{i=1}^n \left(\frac{\ell_i(\hat{\boldsymbol{\theta}}_n^{lw}(s))}{\sum_{j=1}^n \ell_j(\hat{\boldsymbol{\theta}}_n^{lw}(s))} - \frac{1}{n}\right)\left(\ell_i(\boldsymbol{\theta}) - \ell_i(\hat{\boldsymbol{\theta}}_n^{lw}(s))\right) + \frac{1}{n}\sum_{i=1}^n \left(\ell_i(\hat{\boldsymbol{\theta}}_n(t)) - \ell_i(\hat{\boldsymbol{\theta}}_n^{lw}(s))\right)$$

---

[4]One can find empirical evidences of this assumption (the optimal training loss can be zero) in e.g. Zhang et al. (2017) (Figure 1 (a)).

[5]For example, at the initialization, $\hat{L}_n(\hat{\boldsymbol{\theta}}_n(0)) = \hat{L}_n(\boldsymbol{\theta}_0) = \hat{L}_n(\hat{\boldsymbol{\theta}}_n^{lw}(0))$.

$$= -\sum_{i=1}^{n} \underbrace{\left( \frac{\ell_i(\hat{\boldsymbol{\theta}}_n^{\text{lw}}(s))}{\sum_{j=1}^{n} \ell_j(\hat{\boldsymbol{\theta}}_n^{\text{lw}}(s))} - \frac{1}{n} \right) \left( \ell_i(\hat{\boldsymbol{\theta}}_n^{\text{lw}}(s)) - \ell_i(\boldsymbol{\theta}) \right)}_{T_1} + \underbrace{\left( \hat{L}_n(\hat{\boldsymbol{\theta}}_n(t)) - \hat{L}_n(\hat{\boldsymbol{\theta}}_n^{\text{lw}}(s)) \right)}_{T_2}, \quad \text{(A.8)}$$

we analyze $T_1, T_2$ separately.

(i) $T_1$: Note that if $\frac{\ell_i(\hat{\boldsymbol{\theta}}_n^{\text{lw}}(s))}{\sum_{j=1}^{n} \ell_j(\hat{\boldsymbol{\theta}}_n^{\text{lw}}(s))} \leq \frac{1}{n}$ for any $i \in [n]$, we get $\frac{\ell_i(\hat{\boldsymbol{\theta}}_n^{\text{lw}}(s))}{\sum_{j=1}^{n} \ell_j(\hat{\boldsymbol{\theta}}_n^{\text{lw}}(s))} = \frac{1}{n}$ for any $i \in [n]$, which holds in the zero probability and implies the triviality. Let $I^+ := \left\{ i \in [n] : \frac{\ell_i(\hat{\boldsymbol{\theta}}_n^{\text{lw}}(s))}{\sum_{j=1}^{n} \ell_j(\hat{\boldsymbol{\theta}}_n^{\text{lw}}(s))} > \frac{1}{n} \right\} \neq \varnothing$, and $i_{\min}^+ := \arg\min_{i \in I^+} \ell_i(\hat{\boldsymbol{\theta}}_n^{\text{lw}}(s))$, and similarly $I^- := \left\{ i \in [n] : \frac{\ell_i(\hat{\boldsymbol{\theta}}_n^{\text{lw}}(s))}{\sum_{j=1}^{n} \ell_j(\hat{\boldsymbol{\theta}}_n^{\text{lw}}(s))} \leq \frac{1}{n} \right\} \neq \varnothing$, and $i_{\max}^- := \arg\max_{i \in I^-} \ell_i(\hat{\boldsymbol{\theta}}_n^{\text{lw}}(s))$. Obviously, $\ell_{i_{\min}^+}(\hat{\boldsymbol{\theta}}_n^{\text{lw}}(s)) > \frac{1}{n} \sum_{j=1}^{n} \ell_j(\hat{\boldsymbol{\theta}}_n^{\text{lw}}(s)) \geq \ell_{i_{\max}^-}(\hat{\boldsymbol{\theta}}_n^{\text{lw}}(s))$, hence $\delta(s) := \ell_{i_{\min}^+}(\hat{\boldsymbol{\theta}}_n^{\text{lw}}(s)) - \ell_{i_{\max}^-}(\hat{\boldsymbol{\theta}}_n^{\text{lw}}(s)) > 0$ for any $s \geq 0$. Notice that $\hat{L}_n(\boldsymbol{\theta}^*) = 0 \Leftrightarrow \ell_i(\boldsymbol{\theta}^*) = 0, \forall i \in [n]$, we have

$$\begin{aligned}
T_1\big|_{\boldsymbol{\theta}=\boldsymbol{\theta}^*} &= \sum_{i \in I^+} \left( \frac{\ell_i(\hat{\boldsymbol{\theta}}_n^{\text{lw}}(s))}{\sum_{j=1}^{n} \ell_j(\hat{\boldsymbol{\theta}}_n^{\text{lw}}(s))} - \frac{1}{n} \right) \left( \ell_i(\hat{\boldsymbol{\theta}}_n^{\text{lw}}(s)) - \ell_i(\boldsymbol{\theta}^*) \right) \\
&\quad + \sum_{i \in I^-} \left( \frac{\ell_i(\hat{\boldsymbol{\theta}}_n^{\text{lw}}(s))}{\sum_{j=1}^{n} \ell_j(\hat{\boldsymbol{\theta}}_n^{\text{lw}}(s))} - \frac{1}{n} \right) \left( \ell_i(\hat{\boldsymbol{\theta}}_n^{\text{lw}}(s)) - \ell_i(\boldsymbol{\theta}^*) \right) \\
&= \sum_{i \in I^+} \left( \frac{\ell_i(\hat{\boldsymbol{\theta}}_n^{\text{lw}}(s))}{\sum_{j=1}^{n} \ell_j(\hat{\boldsymbol{\theta}}_n^{\text{lw}}(s))} - \frac{1}{n} \right) \ell_i(\hat{\boldsymbol{\theta}}_n^{\text{lw}}(s)) + \sum_{i \in I^-} \left( \frac{\ell_i(\hat{\boldsymbol{\theta}}_n^{\text{lw}}(s))}{\sum_{j=1}^{n} \ell_j(\hat{\boldsymbol{\theta}}_n^{\text{lw}}(s))} - \frac{1}{n} \right) \ell_i(\hat{\boldsymbol{\theta}}_n^{\text{lw}}(s)) \\
&\geq \sum_{i \in I^+} \left( \frac{\ell_i(\hat{\boldsymbol{\theta}}_n^{\text{lw}}(s))}{\sum_{j=1}^{n} \ell_j(\hat{\boldsymbol{\theta}}_n^{\text{lw}}(s))} - \frac{1}{n} \right) \ell_{i_{\min}^+}(\hat{\boldsymbol{\theta}}_n^{\text{lw}}(s)) + \sum_{i \in I^-} \left( \frac{\ell_i(\hat{\boldsymbol{\theta}}_n^{\text{lw}}(s))}{\sum_{j=1}^{n} \ell_j(\hat{\boldsymbol{\theta}}_n^{\text{lw}}(s))} - \frac{1}{n} \right) \ell_{i_{\max}^-}(\hat{\boldsymbol{\theta}}_n^{\text{lw}}(s)) \\
&= \sum_{i \in I^+} \left( \frac{\ell_i(\hat{\boldsymbol{\theta}}_n^{\text{lw}}(s))}{\sum_{j=1}^{n} \ell_j(\hat{\boldsymbol{\theta}}_n^{\text{lw}}(s))} - \frac{1}{n} \right) \left( \ell_{i_{\max}^-}(\hat{\boldsymbol{\theta}}_n^{\text{lw}}(s)) + \delta(s) \right) + \sum_{i \in I^-} \left( \frac{\ell_i(\hat{\boldsymbol{\theta}}_n^{\text{lw}}(s))}{\sum_{j=1}^{n} \ell_j(\hat{\boldsymbol{\theta}}_n^{\text{lw}}(s))} - \frac{1}{n} \right) \ell_{i_{\max}^-}(\hat{\boldsymbol{\theta}}_n^{\text{lw}}(s)) \\
&= \ell_{i_{\max}^-}(\hat{\boldsymbol{\theta}}_n^{\text{lw}}(s)) \sum_{i=1}^{n} \left( \frac{\ell_i(\hat{\boldsymbol{\theta}}_n^{\text{lw}}(s))}{\sum_{j=1}^{n} \ell_j(\hat{\boldsymbol{\theta}}_n^{\text{lw}}(s))} - \frac{1}{n} \right) + \delta(s) \sum_{i \in I^+} \left( \frac{\ell_i(\hat{\boldsymbol{\theta}}_n^{\text{lw}}(s))}{\sum_{j=1}^{n} \ell_j(\hat{\boldsymbol{\theta}}_n^{\text{lw}}(s))} - \frac{1}{n} \right) \\
&= \ell_{i_{\max}^-}(\hat{\boldsymbol{\theta}}_n^{\text{lw}}(s))(1-1) + \Delta(s) = \Delta(s), \quad \text{(A.9)}
\end{aligned}$$

where $\Delta(s) := \delta(s) \sum_{i \in I^+} \left( \frac{\ell_i(\hat{\boldsymbol{\theta}}_n^{\text{lw}}(s))}{\sum_{j=1}^{n} \ell_j(\hat{\boldsymbol{\theta}}_n^{\text{lw}}(s))} - \frac{1}{n} \right) > 0$ for any $s \geq 0$. By continuity, $T_1\big|_{\boldsymbol{\theta}} \geq \Delta(s)/2 > 0$ also holds for any $\boldsymbol{\theta} \approx \boldsymbol{\theta}^*$.

(ii) $T_2$: It measures the difference between losses under the standard and loss-weighted gradient flow. Combining (A.7), (A.8) with (A.9) yields that

$$\begin{aligned}
\frac{d}{ds} \|\hat{\boldsymbol{\theta}}_n^{\text{lw}}(s) - \boldsymbol{\theta}^*\|_2^2 &\leq 2 \left[ \frac{1}{n} \sum_{i=1}^{n} \left( \ell_i(\boldsymbol{\theta}^*) - \ell_i(\hat{\boldsymbol{\theta}}_n(t)) \right) - T_1\big|_{\boldsymbol{\theta}=\boldsymbol{\theta}^*} + T_2 \right] \\
&\leq 2 \left[ \left( \hat{L}_n(\boldsymbol{\theta}^*) - \hat{L}_n(\hat{\boldsymbol{\theta}}_n(t)) \right) - \Delta(s) + \left( \hat{L}_n(\hat{\boldsymbol{\theta}}_n(t)) - \hat{L}_n(\hat{\boldsymbol{\theta}}_n^{\text{lw}}(s)) \right) \right] \\
&= 2 \left[ \left( \hat{L}_n(\boldsymbol{\theta}^*) - \hat{L}_n(\hat{\boldsymbol{\theta}}_n^{\text{lw}}(s)) \right) - \Delta(s) \right], \quad \text{(A.10)}
\end{aligned}$$

which gives

$$\hat{L}_n(\hat{\boldsymbol{\theta}}_n^{\text{lw}}(s)) - \hat{L}_n(\boldsymbol{\theta}^*) \leq -\frac{1}{2} \frac{d}{ds} \|\hat{\boldsymbol{\theta}}_n^{\text{lw}}(s) - \boldsymbol{\theta}^*\|_2^2 - \Delta(s) \quad \text{(A.11)}$$

$$\Rightarrow \int_{s_1}^{s_2} \hat{L}_n(\hat{\boldsymbol{\theta}}_n^{\text{lw}}(s)) ds - (s_2 - s_1) \cdot \hat{L}_n(\boldsymbol{\theta}^*) \leq -\frac{1}{2} \left( \|\hat{\boldsymbol{\theta}}_n^{\text{lw}}(s_2) - \boldsymbol{\theta}^*\|_2^2 - \|\hat{\boldsymbol{\theta}}_n^{\text{lw}}(s_1) - \boldsymbol{\theta}^*\|_2^2 \right) - \int_{s_1}^{s_2} \Delta(s) ds$$

$$\leq \frac{1}{2}\|\hat{\boldsymbol{\theta}}_n^{\mathrm{lw}}(s_1) - \boldsymbol{\theta}^*\|_2^2 - \int_{s_1}^{s_2} \Delta(s)ds \tag{A.12}$$

for any $s_2 > s_1 \geq 0$. That is

$$\frac{1}{s_2 - s_1}\int_{s_1}^{s_2} \hat{L}_n(\hat{\boldsymbol{\theta}}_n^{\mathrm{lw}}(s))ds - \hat{L}_n(\boldsymbol{\theta}^*) \leq \frac{1}{2(s_2 - s_1)}\|\hat{\boldsymbol{\theta}}_n^{\mathrm{lw}}(s_1) - \boldsymbol{\theta}^*\|_2^2 - \frac{1}{s_2 - s_1}\int_{s_1}^{s_2} \Delta(s)ds,$$

or for any $s > 0$,

$$\frac{1}{s}\int_0^s \hat{L}_n(\hat{\boldsymbol{\theta}}_n^{\mathrm{lw}}(s'))ds' - \hat{L}_n(\boldsymbol{\theta}^*) \leq \frac{1}{2s}\|\boldsymbol{\theta}_0 - \boldsymbol{\theta}^*\|_2^2 - \frac{1}{s}\int_0^s \Delta(s')ds'$$

$$< \frac{1}{2s}\|\boldsymbol{\theta}_0 - \boldsymbol{\theta}^*\|_2^2. \tag{A.13}$$

Recall that (A.6) can be rewritten as

$$\frac{d}{dt}\|\hat{\boldsymbol{\theta}}_n(t) - \boldsymbol{\theta}^*\|_2^2 \leq 2\left(\hat{L}_n(\boldsymbol{\theta}^*) - \hat{L}_n(\hat{\boldsymbol{\theta}}_n(t))\right). \tag{A.14}$$

Compared with (A.10), for any $s, t \geq 0$ such that $\hat{L}_n(\hat{\boldsymbol{\theta}}_n(t)) = \hat{L}_n(\hat{\boldsymbol{\theta}}_n^{\mathrm{lw}}(s))$, we have (A.10)'s RHS $< $ (A.14)'s RHS $= -2\hat{L}_n(\hat{\boldsymbol{\theta}}_n(t)) \leq 0$, which implies a sharper convergence rate of the loss-weighted gradient flow (at the same loss level with the standard gradient flow). The proof is completed. □

Proposition A.1 suggests that, under certain regularity conditions, the time-averaged loss of loss-weighted gradient flow converges more than sub-linearly to the global minimum, while the standard gradient flow has the sub-linear convergence. In addition, at the same loss level, the convergence rate of loss-weighted gradient flow is sharper than that of standard gradient flow. This theoretical characterization fundamentally gives chances to learning acceleration by leveraging the loss information in the gradient-based training dynamics.

## A.2    PROOF OF PROPOSITION 2

*Proof.* The decoupled EMAs (3.8) can be rewritten as

$$\boldsymbol{p}(t) \propto \boldsymbol{w}(t) = \beta_1 \boldsymbol{s}(t-1) + (1-\beta_1)\boldsymbol{l}(t),$$
$$\boldsymbol{s}(t) = \beta_2 \boldsymbol{s}(t-1) + (1-\beta_2)\boldsymbol{l}(t), \quad \boldsymbol{s}(0) = \mathbf{1}/n \tag{A.15}$$

In (A.15), let the first equation minus the second, we get

$$\boldsymbol{w}(t) - \boldsymbol{s}(t) = (\beta_2 - \beta_1)(\boldsymbol{l}(t) - \boldsymbol{s}(t-1)). \tag{A.16}$$

The second equation gives

$$\boldsymbol{s}(t) - \boldsymbol{s}(t-1) = (1-\beta_2)(\boldsymbol{l}(t) - \boldsymbol{s}(t-1)). \tag{A.17}$$

Combining (A.16) with (A.17), we have

$$\boldsymbol{w}(t) = \boldsymbol{s}(t) + \frac{\beta_2 - \beta_1}{1 - \beta_2}(\boldsymbol{s}(t) - \boldsymbol{s}(t-1)), \tag{A.18}$$

which proves the first equality.

Expanding the second equation, by induction we get

$$\boldsymbol{s}(t) = \beta_2^t \boldsymbol{s}(0) + (1-\beta_2)\sum_{k=1}^{t} \beta_2^{t-k}\boldsymbol{l}(k), \tag{A.19}$$

hence

$$\boldsymbol{s}(t) - \boldsymbol{s}(t-1) = \beta_2^{t-1}(\beta_2 - 1)\boldsymbol{s}(0) + (1-\beta_2)\left[\sum_{k=1}^{t} \beta_2^{t-k}\boldsymbol{l}(k) - \sum_{k=1}^{t-1} \beta_2^{t-1-k}\boldsymbol{l}(k)\right]$$

$$= -(1-\beta_2)\beta_2^{t-1}\boldsymbol{s}(0) + (1-\beta_2)\left[\beta_2^{t-1}\boldsymbol{l}(1) + \sum_{k=2}^{t} \beta_2^{t-k}\boldsymbol{l}(k) - \sum_{k=1}^{t-1} \beta_2^{t-1-k}\boldsymbol{l}(k)\right]$$

$$= -(1 - \beta_2)\beta_2^{t-1}\boldsymbol{s}(0) + (1 - \beta_2)\left[\beta_2^{t-1}\boldsymbol{l}(1) + \sum_{k=1}^{t-1}\beta_2^{t-1-k}(\boldsymbol{l}(k+1) - \boldsymbol{l}(k))\right]$$

$$\approx (1 - \beta_2)\sum_{k=1}^{t-1}\beta_2^{t-1-k}(\boldsymbol{l}(k+1) - \boldsymbol{l}(k)) \tag{A.20}$$

for relatively large $t$, and the approximation error is exponentially small (due to $\lim_{t\to+\infty}\beta_2^t = 0$ for any $\beta_2 \in (0, 1)$). Combining (A.18), (A.19) and (A.20) yields (3.10), and the proof is completed. $\square$

### A.3 Proof of Proposition 3

*Proof.* The problem (3.12) can be solved in an alternative gradient descent-ascent manner:

$$\boldsymbol{\theta}(t + 1) = \boldsymbol{\theta}(t) - \eta_t^{\boldsymbol{\theta}}\sum_{i=1}^{n}p_i(t)\nabla_{\boldsymbol{\theta}}\ell_i(\boldsymbol{\theta}(t)),$$

$$w_i(t + 1) = w_i(t) + \eta_t^{\boldsymbol{w}}(\ell_i(\boldsymbol{\theta}(t+1)) - \ell_i^{\mathrm{ref}}), \quad p_i(t) = \frac{w_i(t)}{\sum_j w_j(t)}. \tag{A.21}$$

Decoupled EMAs (3.8) update the weights as

$$w_i(t + 1) = w_i(t) + (1 - \beta_1)(\ell_i(\boldsymbol{\theta}(t+1)) - \ell_i(\boldsymbol{\theta}(t))) + \beta_1(s_i(t) - s_i(t - 1)). \tag{A.22}$$

By (A.19), we get

$$s_i(t) - s_i(t - 1) = -(1 - \beta_2)\beta_2^{t-1}s_i(0) - (1 - \beta_2)^2\sum_{k=1}^{t-1}\beta_2^{t-1-k}\ell_i(\boldsymbol{\theta}(k)) + (1 - \beta_2)\ell_i(\boldsymbol{\theta}(t)),$$

hence

$$w_i(t + 1) = w_i(t) + (1 - \beta_1)(\ell_i(\boldsymbol{\theta}(t+1)) - \ell_i(\boldsymbol{\theta}(t))) - \beta_1(1 - \beta_2)\beta_2^{t-1}s_i(0)$$

$$- \beta_1(1 - \beta_2)^2\sum_{k=1}^{t-1}\beta_2^{t-1-k}\ell_i(\boldsymbol{\theta}(k)) + \beta_1(1 - \beta_2)\ell_i(\boldsymbol{\theta}(t)). \tag{A.23}$$

Let

$$\ell_i^{\mathrm{ref}} = \frac{1 - 2\beta_1 + \beta_1\beta_2}{1 - \beta_1}\ell_i(\boldsymbol{\theta}(t)) + \frac{\beta_1(1 - \beta_2)^2}{1 - \beta_1}\sum_{k=1}^{t-1}\beta_2^{t-1-k}\ell_i(\boldsymbol{\theta}(k)) + \frac{\beta_1(1 - \beta_2)\beta_2^{t-1}}{1 - \beta_1}s_i(0), \tag{A.24}$$

then we have

$$w_i(t + 1) = w_i(t) + (1 - \beta_1)(\ell_i(\boldsymbol{\theta}(t+1)) - \ell_i^{\mathrm{ref}}), \tag{A.25}$$

which coincides with the update formula (A.21) with $\eta_t^{\boldsymbol{w}} = 1 - \beta_1$. The proof is completed. $\square$

## B More Details of Experiments

In this section, we present further experimental details. We run all the experiments with one NVIDIA A100 (80GB) with the mixed-precision training. All the algorithms are implemented based on PyTorch (Paszke et al., 2019) and Timm (Wightman et al., 2019). For InfoBatch, our implementation is adapted from Qin et al. (2024).

### B.1 Experiments on CIFAR Datasets

For the experiments on the CIFAR-10/100 datasets, we use the SGD optimizer with the momentum 0.9 and weight decay $5 \times 10^{-4}$. We apply the OneCycle scheduler (Smith & Topin, 2019) with the cosine annealing. For CIFAR-10, the maximal learning rate is 0.2 for the baseline and *set* level selection methods, while 0.05 for *batch* level selection methods due to larger variances of stochastic gradients and 0.08 for ESWP. For CIAFR-100 trained with ResNet-18/50, the maximal learning rates for all the sampling methods are 0.05/0.2, following Qin et al. (2024).

For the experiments under light/heavy label noises, we uniformly select 10%/40% samples in the whole dataset and assign them wrong labels with uniform probabilities for *uniform* noises or a certain label for *flip* noises (Ghosh & Lan, 2021, Section 2).

## B.2 EXPERIMENTS OF FINE-TUNING

**ALBERT.** Following the setup in Xie et al. (2023b) (Table 8), we use the AdamW optimizer and the polynomial decay scheduler with the warm up.

**Vision Transformer.** We finetune the ViT-Large model on the ImageNet-1K dataset with a meta-batch size $B = 256$ for 10 epochs, using the Adam optimizer with the OneCycle scheduler (Smith & Topin, 2019) with the cosine annealing and a maximal learning rate of $2 \times 10^{-5}$.

## B.3 ADDITIONAL PLOTS

Following Mindermann et al. (2022), we plot the test accuracy versus the number of samples used for back-propagation (BP) for Baseline and ES(WP) in Figure 6. It is clear that ES(WP) can significantly reduce the BP costs and thus improves the learning efficiency.

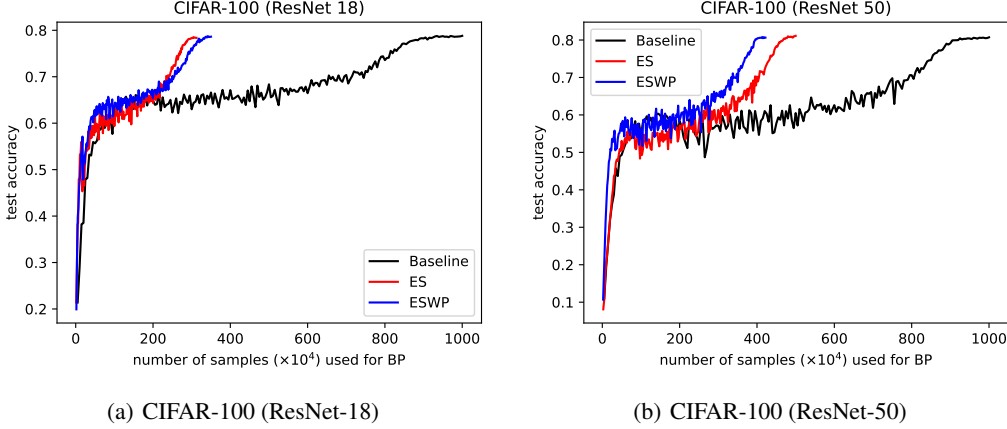

(a) CIFAR-100 (ResNet-18)      (b) CIFAR-100 (ResNet-50)

Figure 6: Learning dynamics of different data selection methods: Test accuracy versus the number of samples used for back-propagation.

