# OpenReview forum: "Data-Efficient Training by Evolved Sampling"
_ICLR.cc/2025/Conference — Submitted to ICLR 2025_

### Official Review · Reviewer_4oRD · 2024-10-27

**Soundness:** 3
**Presentation:** 4
**Contribution:** 2
**Rating:** 8
**Confidence:** 4

**Summary:**

This paper functions as a well-thought-out "momentum optimizer" in the data space. Instead of considering the presentation of data as fixed as in SGD, we take a more expansive view and think of the data space as another component of the model to optimize.

The work is somewhat novel in the large model training space.

**Strengths:**

The paper builds upon good theoretical foundations.

The paper well cites related work and the literature that leads to this contribution.

The paper creates an efficient heuristic based approach to solve a practical problem which rests on the previous theoretical contributions.

The paper well considers ablation studies and robustness studies.

The paper's theoretical arguments are well constructed.

**Weaknesses:**

This should be better justified: This can be inefficient since different samples may have varied importance. Can you look at the influence functions or coresets literature?

This statement needs to be better motivated and explained, why is evolved sampling "natural?"
In general machine learning tasks, the typical behaviors of loss curves often appear decent trends
overall, but can oscillate meanwhile due to certain noises. This introduces the sensitivity or instability
issue of the sampling scheme (3.6). A natural smoothing operation is to use the exponential moving
average (EMA) of losses

The proof presentations are somewhat lacking. It's difficult for me to quickly match up concepts from the optimization literature to some of the theoretical arguments made, for example, the EMA to the minimax problem.

It may be worthwhile in explaining this better with regards to the control theory literature, specifically, control theory also deals with oscillations and rectifies them in similar manners:

Decoupled EMA. To sufficiently leverage the loss dynamics in a more robust sense, we propose to
calculate the sampling probability as
pi(t) ∝ wi(t) = β1si(t − 1) + (1 − β1)ℓi(θ(t)),
si(t) = β2si(t − 1) + (1 − β2)ℓi(θ(t)), si(0) = 1/n (3.8)
with β1, β2 ∈ [0, 1] as two hyper-parameters. Here, the intermediate series {si(t)}t∈N, updated in
the EMA scheme, is also referred as the score (for the i-th sample). The scheme (3.8) is the so-called
decoupled EMA,
2 which reduces to (3.7) when β1 = β2 = β. In Figure 1, it is shown by the red curve
and appears an “interpolation” between the original loss and single EMA: When losses oscillate,
the decoupled EMA reacts moderately by not only capturing detailed dynamics of losses, but also
remaining necessary robustness , exhibiting the flexibility to trade-off (by tuning two betas).
Intuitively, by setting (β1, β2) → (0+, 1
−), we are able to exploit the long-term historical information
along the training (via β2), while focusing on the importance of current losses (via β1) and thus can
get the best of both world. This simple and elegant design turns out to be surprisingly beneficial in
practice, which is further verified in numerous experiments in Section 4.


This should really be better explained. Again, this paper is moving into the "total optimization landscape" where both data and model parameters are considered components of the system to be optimized. It's not immediately clear whether this is a consequence of the problem the authors were solving, or the key insight that led to the approach.

(ii) ES to solve a DRO problem. From another perspective, ES can be also reformulated as a
solution to the minimax problem...

**Questions:**

Can the key idea of the paper: optimization of the data space, be more cohesively or clearly presented? Currently, it's still difficult to understand the key idea of the paper without significant theoretical and literature knowledge.

---

> ### Author Response · Authors · 2024-11-27
> **Response to Reviewer 4oRD**
>
> >**Q1**: This should be better justified: This can be inefficient since different samples may have varied importance. Can you look at the influence functions or coresets literature?
>
> **A1**: For coresets:
> - We have cited related coresets literature in Section 2 (see the "Static sampling" paragraph, Line 118-122 in the original manuscript). As we have discussed, static sampling methods require *extra* training, leading to considerable costs in both computation and memory.
>
> For influence functions:
> - The definition of influence functions involves calculations of high-dimensional gradients, Hessians and their inverses, whose computation overheads are considerably significant.
>
> >**Q2**: This statement needs to be better motivated and explained, why is evolved sampling "natural?" In general machine learning tasks, the typical behaviors of loss curves often appear decent trends overall, but can oscillate meanwhile due to certain noises. This introduces the sensitivity or instability issue of the sampling scheme (3.6). A natural smoothing operation is to use the exponential moving average (EMA) of losses.
>
> **A2**: Sorry for the wrong description. We actually mean "commonly-used" here. We have updated this in the revised version.
>
> >**Q3**: The proof presentations are somewhat lacking. It's difficult for me to quickly match up concepts from the optimization literature to some of the theoretical arguments made, for example, the EMA to the minimax problem.
>
> **A3**: Te proof of Proposition 3 is deferred to Appendix A.3. Proposition 3 means that, solving the minimax optimization problem (3.12) with gradient-based iterations is formally equivalent to gradient descent iterations integrated with the decoupled EMA sampling. This provides a novel perspective to understand EMA-based  data selection methods, which has not been discussed in the optimization literature to the best of our knowledge.
>
> >**Q4**: It may be worthwhile in explaining this better with regards to the control theory literature, specifically, control theory also deals with oscillations and rectifies them in similar manners.
>
> **A4**: Thanks for your insightful suggestions. Yes, control theory in general deals with behaviors of dynamical systems, and it is straightforward to formulate the training with data selection as certain constrained optimization problems, and possibly derive corresponding necessary conditions with tools in optimal control (e.g. Pontryagin’s maximum principle). However, it seems that this direction can suffer from significant computation loads due to additionally involved calculations of gradients and Hessians. We would appreciate if you can suggest more related references.

---

> ### Author Response · Authors · 2024-11-27
> **Response to Reviewer 4oRD (continue)**
>
> >**Q5**: "Decoupled EMA. To sufficiently leverage the loss dynamics in a more robust sense, we propose to calculate the sampling probability as $p_i(t) \propto w_i(t) = \beta_1 s_i(t-1)+(1-\beta_1)\ell_i(\theta(t))$, $s_i(t) = \beta_2 s_i(t-1)+(1-\beta_2)\ell_i(\theta(t))$, $s_i(0) = 1/n$ with $\beta_1, \beta_2 \in [0,1]$ as two hyper-parameters. Here, the intermediate series $\{s_i(t)\}_{t\in\mathbb{N}}$, updated in the EMA scheme, is also referred as the score (for the i-th sample). The scheme (3.8) is the so-called decoupled EMA, which reduces to (3.7) when $\beta_1=\beta_2=\beta$. In Figure 1, it is shown by the red curve and appears an “interpolation” between the original loss and single EMA: When losses oscillate, the decoupled EMA reacts moderately by not only capturing detailed dynamics of losses, but also remaining necessary robustness , exhibiting the flexibility to trade-off (by tuning two betas). Intuitively, by setting $(\beta_1, \beta_2) \to (0^+, 1^-)$, we are able to exploit the long-term historical information along the training (via  $\beta_2$), while focusing on the importance of current losses (via $\beta_1$) and thus can get the best of both world. This simple and elegant design turns out to be surprisingly beneficial in practice, which is further verified in numerous experiments in Section 4."
> This should really be better explained. Again, this paper is moving into the "total optimization landscape" where both data and model parameters are considered components of the system to be optimized. It's not immediately clear whether this is a consequence of the problem the authors were solving, or the key insight that led to the approach.
>
> **A5**: For statements' explainations:
> - Relations of EMAs: Figure 1 illustrates the comparisons among dynamics of EMAs. It is shown that the decoupled EMA acts as an *interpolation* between the loss re-weighting and standard EMA: When losses of certain samples oscillate, the decoupled EMA reacts moderately by not only capturing detailed dynamics of losses (which is ignored by standard EMA due to its over-smoothing effect), but also remaining certain robustness (while loss re-weighting is quite sensitive to loss variations). The decoupled EMA is also flexible to trade-off between these two regimes (details and smoothing) by tuning two betas (left $\to$ middle $\to$ right in Figure 1).
> - Effect of $(\beta_1, \beta_2)$:
>     - Since $p_i(t) \propto w_i(t) = \beta_1 s_i(t-1)+(1-\beta_1)\ell_i(\theta(t))$, $\beta_1 \in [0,1]$, it is obvious that smaller $\beta_1$ gives a larger coefficient of the current loss $\ell_i(\theta(t))$, hence we are focusing on the importance of current losses by setting $\beta_1 \to 0^+$.
>     - Since $s_i(t) = \beta_2 s_i(t-1)+(1-\beta_2)\ell_i(\theta(t))$, $\beta_2 \in [0,1]$, it is obvious that larger $\beta_2$ gives a larger coefficient of the historical score $s_i(t-1)$, hence we are focusing on the importance of historical weights by setting $\beta_2 \to 1^-$.
>
> For "total optimization landscape":
> - We agree with this viewpoint, but it can be quite general in terms of induced formulations. In our opinion, specific realizations of "total optimization landscape" should be simple in the sense to introduce additional computation as light as possible. We currently view data selection with high-order loss-based re-weighting (i.e. decoupled EMA) as an economical candidate (see reasons in the 3rd and 5th sub-point of the 2nd point in **A1** in **Response to Reviewer R7ur**).
>
> >**Q6**: (ii) ES to solve a DRO problem. From another perspective, ES can be also reformulated as a solution to the minimax problem...
>
> **A6**: Please refer to **A3** for details.
>
> >**Q7**: Can the key idea of the paper: optimization of the data space, be more cohesively or clearly presented? Currently, it's still difficult to understand the key idea of the paper without significant theoretical and literature knowledge.
>
> **A7**: For the discussion of key ideas of this paper, you can refer to the last point in **A5** for details. For the writing part, although it is recognized by other reviewers, say Reviewer G9Hm: "The paper is clearly structured, with well-organized sections..."; Reviewer R7ur: "Overall the paper is clearly and concisely written", we are certainly open to more specific suggestions.

---

> > ### Comment · Reviewer_4oRD · 2024-11-27
> > **Response to author**
> >
> > I am convinced of the merits of the paper. I will be raising my score.
> >
> > The authors should consider the feedback of other reviewers as well and think about improving the presentation, as well as the validation to put forward a more persuasive argument.

---

> > > ### Author Response · Authors · 2024-12-02
> > > **Response to Reviewer 4oRD**
> > >
> > > Thank you for your reply! We are happy to see that our response addressed your concerns. Thanks again for your valuable feedback to help us improve our paper and for increasing your rating.

---

### Official Review · Reviewer_R7ur · 2024-11-01

**Soundness:** 2
**Presentation:** 3
**Contribution:** 2
**Rating:** 5
**Confidence:** 3

**Summary:**

The paper introduces a novel framework called Evolved Sampling (ES) (and with Pruning ES-WP) aimed at enhancing data efficiency in machine learning. The authors propose a dynamic sampling method that selects informative data samples based on the evolution of losses during training. This approach aims to reduce backpropagation time while maintaining model performance across various architectures (ResNet, ViT, ALBERT) and datasets (CIFAR, ImageNet, GLUE). Key contributions include: (i) Dynamic Sampling: ES utilizes historical and current loss differences to inform data selection, allowing for batch-level sampling without the need for pre-trained models. (ii)Efficiency Gains: The method achieves up to 40% reduction in wall-clock time during training and shows improved accuracy (approximately 20%) in scenarios with noisy labels; and (iii) Theoretical Justifications: The authors provide theoretical insights into how their method alleviates loss oscillations and can be viewed through the lens of distributionally robust optimization.

**Strengths:**

### Originality:

The main proposition lies in the recursive definition of an Exponentially Moving Average over the losses of individual examples to deselect them from the training process to gain speedups and improved; i.e. stable, learning dynamics. The single-level EMA itself is a well-known approach that is applied to this setting with a recursive definition. The other techniques, i.e. annealing and pruning, are mere adaptations from prior work and are only a minor contribution to the originality. The bridge between batch and set-level data selection, which their method allows them to do is a nice feature, but not the main contribution. The theoretic analysis is interesting overall. But insights like decoupled EMA is in fact a convolution over hyperparameters’ powers of historical losses – so their results are not really surprising.

### Quality:

Quite a few experimental issues are present, which I will detail in the weaknesses section.

### Clarity:

Overall the paper is clearly and concisely written. With the main exception of when exactly we are collecting the loss values of pruned examples; which might bias the calculation of their weight.

### Significance:

The efficiency of modern machine learning algorithms and neural networks is a great issue, as it results in huge energy demand. Reducing the footprint is a critical point. One angle of attack pursued in this paper is being selective about the order and the subset of consumed examples. This is indeed an important and interesting avenue.

**Weaknesses:**

Besides the weak overall originality, my main criticism is connected to the empirical evaluation:

The necessity for a burn-in period, where standard training must occur to initialize the loss adequately before applying the Exponential Moving Average (EMA) scheme, points to a limitation in the approach. This dependency on a specific loss initialization suggests that the method might not be entirely robust across various starting conditions. It would benefit the study to explore a more systematic ablation of this burn-in period as a hyperparameter. Additionally, understanding whether variations in the burn-in length affect performance could provide insight into the model's dependency on initialization stability and might even reveal opportunities to shorten or eliminate this requirement.

Another area where clarity is needed is the reporting of statistical measures. The number of seeds used for evaluation and averaging remains unspecified, and no standard deviations are provided. This omission raises questions about whether noise rather than true performance gains might influence observed differences in performance between the proposed method and baseline competitors. Including standard deviations would allow readers to assess the consistency of the results, providing a clearer understanding of the variability in performance.

The use of wall-clock time as a measure of speedup also presents challenges. Since wall-clock time is influenced by multiple factors, including the specific point of reference and the extent to which reference performance is met or exceeded, this metric is not straightforward. No details are provided on the variability of wall-clock measurements, which could make these results more challenging to interpret. An additional, complementary metric—such as the number of examples seen (similar to token counts in LLM training)—could yield a more direct and comparable measurement of processing efficiency, especially since the baseline approach involves higher computational requirements.

Regarding robustness to label noise, Figure 3a indicates that while the method outperforms the baseline, the speedup advantage is lost under noisy conditions. This finding implies that the method may benefit from integrating the baseline up to its peak performance before switching to the proposed scheme. Such a hybrid approach could potentially leverage the best of both methods, maintaining efficiency without sacrificing performance under challenging conditions.

In Figure 3b, the gradients under comparison lack clarity. It is uncertain whether the gradients displayed encompass all examples (both corrupted and uncorrupted), necessitating additional forward passes and potentially affecting wall-clock measurements, or if the results only include corrupted examples selected by the method. The latter case would introduce a selection bias, affecting the integrity of the reported results. A more informative and balanced approach would be to calculate the proportion of non-informative examples selected per epoch, providing a relative measure of their influence on learning. This would give a clearer picture of how these less useful samples affect training efficiency and could allow for more balanced comparisons.

In Table 5, the ground-truth results are presented without a corresponding baseline for corruption-free performance. Including such a baseline would clarify the upper bound achievable in the absence of noise, providing a benchmark against which the "superior" performance in noisy conditions could be assessed.


Further minor Issues:

* Ablations:
  * choices of \beta. The presented heatmap tables are way too broad. I suggest using some Sobol or Latin Hypercube design and then reporting the heat surfaces. This way, we get a far more fine-grained perspective on the hyperparameters’ behavior.
  * Pruning is not ablated
* The notation 0^+ and 1^- should probably be introduced or replaced by intervals (0, 1) instead of [0, 1]
* The notation is at times slightly overburdened (e.g. the additional vector notation in 320), instead of just writing the actual values in there directly.

**Questions:**

I would like to get a clarification regarding Eq. 3.8. We have access to the current loss of an example to decide whether or not we want to sample it for that epoch. I interpret this as doing the forward pass on an example that we later deselect to be part of the backward pass calculation. This means that we still maintain the gradient of that example until we deselect it. The main cost saved then is the amount of bwd passes. In Algorithm 1, the necessity for forward passes seems to be mitigated in Line 284 at least during the pruning by taking the historically weighed score s instead of the weight function. This seemingly implies that to select examples, only historic losses are considered. But this poses yet another question: How do we adjust an example’s loss if the example is no longer selected? Because then we yet again will need a fwd pass and we could have calculated the full weight. This seems to be what is done in 289; i.e. only the loss over the batch examples is calculated. The only thing to mitigate the issue of disregarding bad losses (almost) completely is in Remark 1 and discounting the existing values. Either way, this introduces non-trivial and dead-lock-ish dynamics I would like to see investigated.

---

> ### Author Response · Authors · 2024-11-27
> **Response to Reviewer R7ur**
>
> >**Q1**: The main proposition lies in the recursive definition of an Exponentially Moving Average over the losses of individual examples to deselect them from the training process to gain speedups and improved; i.e. stable, learning dynamics. The single-level EMA itself is a well-known approach that is applied to this setting with a recursive definition. The other techniques, i.e. annealing and pruning, are mere adaptations from prior work and are only a minor contribution to the originality. The bridge between batch and set-level data selection, which their method allows them to do is a nice feature, but not the main contribution. The theoretic analysis is interesting overall. But insights like decoupled EMA is in fact a convolution over hyperparameters’ powers of historical losses – so their results are not really surprising.
>
> **A1**: Regarding the originality, this work is motivated by mainly loss re-weighting and EMAs. However, the present work differentiates from these existing concepts with at least three points as follows:
> - Loss re-weighting: Although the loss re-weighting is one of our motivations, its effectiveness was only numerically verified in applications and no theoretical characterizations were provided before. In this work, we first mathematically prove the convergence acceleration achieved by loss re-weighting (Proposition 1), which is certainly novel.
> - EMAs: Although the standard EMA has been adopted in data selection (e.g. UCB ([1])), its practical performance is not that satisfying (e.g. [1]). In this work, we propose a new EMA scheme (decoupled EMA), acting as a non-trivial extension in the following sense:
>     - In formulation, the decoupled EMA "space" contains former loss re-weighting based data selection methods. It is easy to see that decoupled EMA reduces to standard EMA by setting $\beta_1=\beta_2=\beta$, and further reduces to basic loss re-weighting and uniform sampling by setting $\beta=0$ and $\beta=1$, respectively.
>     - Intuitively, we illustrate the comparisons among dynamics of EMAs (Figure 1). It is shown that the decoupled EMA acts as an interpolation between the loss re-weighting and standard EMA: When losses oscillate, the decoupled EMA reacts moderately by not only capturing detailed dynamics of losses (which is ignored by standard EMA due to its over-smoothing effect), but also remaining certain robustness (while loss re-weighting is quite sensitive to loss variations). The decoupled EMA is also flexible to trade-off between these two regimes (details and smoothing) by tuning two betas (left $\to$ middle $\to$ right in Figure 1).
>     - In theory, we mathematically prove the novel Proposition 2 to demonstrate that the decoupled EMA is in fact a **first-order** modification of standard EMA, resulting in the flexibility to balance between losses and their **differences**. This potentially gives chances to better leverage the loss dynamics for data selection. Moreover, this first-order modification is *implicitly* introduced, meaning that only quantities involving losses are evaluated, without significant overheads as former first-order data selection methods based on gradients.
>     (This point is also recognized by other reviewers, say Reviewer G9Hm: "Novelty - The paper introduces decoupled exponential moving averages, which leverage *first-order* loss differences for more stable and robust sampling, effectively combining ideas from loss and gradient-based sampling with robust optimization principles.")
>     - Empirically, we observe non-trivial improvements in all experiments presented in Section 4, especially for datasets with noisy labels (*limited* investigations on this setting in former data selection references). Particularly, for the superiority of decoupled EMA over standard EMA, the theoretical insight derived in Proposition 2 is also numerically verified in Table 2, 3 and 5, where ES(WP) outperforms UCB, and abaltions in Table 6, where Loss + DE outperforms Loss + E for multiple architectures and datasets.
>     - Combining theoretical justifications and numerical verifications gives that the *general* convolutional forms are *not* the keys, and introducing additional loss *differences* as the high-order information of loss variations is novel, effective and non-trivial due to its efficiency (recall that this first-order modification is *implicitly* introduced, meaning that only quantities involving losses are evaluated, without significant overheads as former first-order data selection methods based on gradients).
> - Additionally, the proposed method performs data selection in both the batch and set (epoch) level, which is also novel compared to former data selection literature (Table 1). This leads to more aggressive data pruning with better training accelerations and efficiency in appications.

---

> ### Author Response · Authors · 2024-11-27
> **Response to Reviewer R7ur (continue)**
>
> >**Q2**: Overall the paper is clearly and concisely written. With the main exception of when exactly we are collecting the loss values of pruned examples; which might bias the calculation of their weight.
>
> **A2**: For the set level selection (line 282-284), we do not compute the current loss as line 289-290 do (for batch level selection). The main reason is to avoid unnecessary computation overheads. For the bias of weights, firstly, we use sampling instead of ranking to select training data, as is stated in Remark 1,  indicating that even samples with small scores can be still selected probably into $\mathcal{D}_e$.
>
> Moreover, we elaborate the intution behind ESWP as follows:
> (1) The annealing strategy at the initial stage is conducive to data samples towards similar score scales at the first few epochs.
> (2) Suppose the loss value during training has a decaying trend, the score is also likely to decrease.
> (3) If a sample $z_i$ has a relatively small score initially and thus not been selected, its score will remain the same, while the scores of other selected samples are basically decreasing.
> (4) In this way, at some later stages, the score of $z_i$ will become relatively large compared to others, and hence $z_i$ is likely to be selected. Otherwise, the score of $z_i$ is too small, and $z_i$ can be regarded as unimportant or well-fitted sample.
>
> >**Q3**: The necessity for a burn-in period, where standard training must occur to initialize the loss adequately before applying the Exponential Moving Average (EMA) scheme, points to a limitation in the approach. This dependency on a specific loss initialization suggests that the method might not be entirely robust across various starting conditions. It would benefit the study to explore a more systematic ablation of this burn-in period as a hyperparameter. Additionally, understanding whether variations in the burn-in length affect performance could provide insight into the model's dependency on initialization stability and might even reveal opportunities to shorten or eliminate this requirement.
>
> **A3**: Regarding the annealing technique, we respond by two points:
> - Introducing the annealing leads to a hybrid data selection mechanism, which can be viewed as a homotopy or interpolation regime: When the annealing ratio is $\text{ar}:=E_a/E=0$, it is pure ES(WP); when $\text{ar}=0.5$, it reduces to the baseline without any data selection. Therefore, there exist optimal annealing ratio values $\text{ar}^*$ potentially better than both methods. In this work, we set a default $\text{ar}=5$\% for all experiments, resulting in a burn-in period that is far from adequate.
> - **We further perform ablation studies on the annealing ratio $\text{ar}$ to demonstrate its impacts on the overall performance.**
>
>     - Table 5: The effect of annealing ratios in ES (ResNet-18, CIFAR-100).
>         |$\text{ar}$ |$0.0$|$0.05$|$0.075$|
>         |----|----|----|----|
>         |accuracy (\%)|$78.60$|$78.79$|$78.32$|
>
> >**Q4**: Another area where clarity is needed is the reporting of statistical measures. The number of seeds used for evaluation and averaging remains unspecified, and no standard deviations are provided. This omission raises questions about whether noise rather than true performance gains might influence observed differences in performance between the proposed method and baseline competitors. Including standard deviations would allow readers to assess the consistency of the results, providing a clearer understanding of the variability in performance.
>
> **A4**: Regarding the statistical measures, we respond by four points:
> - We *do* run multiple seeds and take the average. In Line 384-385 in the original manuscript: "All the reported results are evaluated on the average of 2-4 independent random trials." All the methods have similar standard deviations, which are between 0.1 and 0.2 in Table 2 and Table 3. We omit this metric for concise presentations.
>
> In fact:
> - For tasks on clean datasets, the gaps of learning accuracies are not that significant, and the acceleration is our main focus. One can observe that in all experiments, the gaps of reduced training time among data selection methods are not marginal, not likely caused by noises (Table 2, 3, 4).
> - For tasks on datasets with noisy labels, the gaps of learning accuracies are significantly large, impossibly caused by noises in training (Table 5, Figure 3(a)).

---

> ### Author Response · Authors · 2024-11-27
> **Response to Reviewer R7ur  (continue)**
>
> >**Q5**: The use of wall-clock time as a measure of speedup also presents challenges. Since wall-clock time is influenced by multiple factors, including the specific point of reference and the extent to which reference performance is met or exceeded, this metric is not straightforward. No details are provided on the variability of wall-clock measurements, which could make these results more challenging to interpret. An additional, complementary metric—such as the number of examples seen (similar to token counts in LLM training)—could yield a more direct and comparable measurement of processing efficiency, especially since the baseline approach involves higher computational requirements.
>
> **A5**: The experiments in the original manuscript mainly follow InfoBatch ([2]; ICLR 2024 Oral) to evaluate data selection methods running with *fixed budgets in epochs*. **To avoid possible interpretation issues, as suggested, we further add plots regarding the global learning dynamics, i.e. test accuracies versus the number of samples used for back-propagation. See Section B.3 (Figure 6) in the revised manuscript for details.**
>
> >**Q6**: Regarding robustness to label noise, Figure 3(a) indicates that while the method outperforms the baseline, the speedup advantage is lost under noisy conditions. This finding implies that the method may benefit from integrating the baseline up to its peak performance before switching to the proposed scheme. Such a hybrid approach could potentially leverage the best of both methods, maintaining efficiency without sacrificing performance under challenging conditions.
>
> **A6**: This is a sharp observation and promising insight. In fact, as is pointed out in **A3**, we have already partially realized this hybrid approach by tuning the annealing ratio $\text{ar}$. The remaining question is how to decide the moment to switch. As an outlook, the most direct way is to monitor the overall performance and try to identify where saturations occur. More advanced techniques may be inspired by [3]: When indexes of mislabeled data samples are known and not dominant in numbers, one can turn to monitor training errors on mislabeled samples, and decide to switch when these errors are maximized. This is based on the empirical findings that gradient descent is biased to fit the clean data first during initial phases of training, and then fit noises. Note that this phenomenon is also observed in Figure 3, where the learning accuracy of "Baseline" first increases and then deceases (Figure 3(a)), with the corresponding noisy gradient ratio first deceasing (bounded) and then increasing (Figure 3(b)).
>
> >**Q7**: In Figure 3(b), the gradients under comparison lack clarity. It is uncertain whether the gradients displayed encompass all examples (both corrupted and uncorrupted), necessitating additional forward passes and potentially affecting wall-clock measurements, or if the results only include corrupted examples selected by the method. The latter case would introduce a selection bias, affecting the integrity of the reported results. A more informative and balanced approach would be to calculate the proportion of non-informative examples selected per epoch, providing a relative measure of their influence on learning. This would give a clearer picture of how these less useful samples affect training efficiency and could allow for more balanced comparisons.
>
> **A7**: We first clarify that Figure 3(b) is plotted to intepret Figure 3(a), not as a part of the proposed method. For further clarifications and discussions, you can refer to **A5** in **Response to Reviewer M8jH** for details.
>
> >**Q8**: In Table 5, the ground-truth results are presented without a corresponding baseline for corruption-free performance. Including such a baseline would clarify the upper bound achievable in the absence of noise, providing a benchmark against which the "superior" performance in noisy conditions could be assessed.
>
> **A8**: Thanks for your suggestion. We have added the corresponding results into Table 5 in the revised version. In fact, these results are exactly the second column of Table 2.

---

> ### Author Response · Authors · 2024-11-27
> **Response to Reviewer R7ur (continue)**
>
> >**Q9**:  Ablations of betas. The presented heatmap tables are way too broad. I suggest using some Sobol or Latin Hypercube design and then reporting the heat surfaces. This way, we get a far more fine-grained perspective on the hyperparameters’ behavior.
>
> **A9**: We plot Figure 5 to convey the main message that default configurations of hyperparameters $(\beta_1, \beta_2)$ are basically okay: The default betas are consistently validated to be roughly (locally) optimal in small-scale models and datasets, and their *superior* effectiveness and efficiency remain in other experiments under different settings (e.g. noisy supervision, Table 5 and Figure 3) and with larger scales (e.g. Table 3). **We also extend Figure 5 for denser betas around the default values to verify its (local) optimality.**
>
> - Table 3: Test accuracies (\%) for different betas in ES (ResNet-18, CIFAR-100).
>     |$\beta_2$ \ $\beta_1$|$0.15$|$0.2$|$0.25$
>     |----|----|----|----|
>     |$0.95$|$78.5$|$78.8$|$78.1$
>     |$0.9$|$78.4$|$\textbf{78.8}$|$78.6$
>     |$0.85$|$78.3$|$78.4$|$78.3$
>
> >**Q10**: Pruning is not ablated.
>
> **A10**: By combining both the batch and set level data selection, we aim to achieve more aggressive data pruning. **We further perform ablation studies on the pruning ratio $\text{pr}$ to demonstrate its impacts on the overall performance.**
>
> - Table 4: The effect of pruning ratios in ESWP (ResNet-18, CIFAR-10).
>     |$\text{pr}$|$0.1$|$0.2$|$0.3$|$0.4$|
>     |----|----|----|----|----|
>     |accuracy (\%)|$95.3$|$95.3$|$95.2$|$94.9$
>
> >**Q11**: The notation $0^+$ and $1^-$ should probably be introduced or replaced by intervals $(0, 1)$ instead of $[0, 1]$.
>
> **A11**: These notations represent the single-side limit. Roughly speaking, $(\beta_1, \beta_2) \to (0^+, 1^-)$ means that $\beta_1$ is close to $0$ with $\beta_1>0$, and $\beta_2$ is close to $1$ with $\beta_2<1$.
>
> >**Q12**: The notation is at times slightly overburdened (e.g. the additional vector notation in 320), instead of just writing the actual values in there directly.
>
> **A12**: Thanks for your suggestion. We have rewritten the suggested contents in the revised version.
>
> >**Q13**: I would like to get a clarification regarding Eq. 3.8. We have access to the current loss of an example to decide whether or not we want to sample it for that epoch. I interpret this as doing the forward pass on an example that we later deselect to be part of the backward pass calculation. This means that we still maintain the gradient of that example until we deselect it. The main cost saved then is the amount of bwd passes. In Algorithm 1, the necessity for forward passes seems to be mitigated in Line 284 at least during the pruning by taking the historically weighed score s instead of the weight function. This seemingly implies that to select examples, only historic losses are considered. But this poses yet another question: How do we adjust an example’s loss if the example is no longer selected? Because then we yet again will need a fwd pass and we could have calculated the full weight. This seems to be what is done in 289; i.e. only the loss over the batch examples is calculated. The only thing to mitigate the issue of disregarding bad losses (almost) completely is in Remark 1 and discounting the existing values. Either way, this introduces non-trivial and dead-lock-ish dynamics I would like to see investigated.
>
> **A13**: Please refer to **A2** for details.
>
> **References**
>
> [1] Ravi Raju, Kyle Daruwalla, and Mikko Lipasti. Accelerating deep learning with dynamic data pruning. *arXiv preprint arXiv:2111.12621*, 2021.
>
> [2] Ziheng Qin, Kai Wang, Zangwei Zheng, Jianyang Gu, Xiangyu Peng, Zhaopan Xu, Daquan Zhou, Lei Shang, Baigui Sun, Xuansong Xie, and Yang You. InfoBatch: Lossless training speed up by unbiased dynamic data pruning. In *International Conference on Learning Representations*, 2024.
>
> [3] Saurabh Garg, Sivaraman Balakrishnan, J. Zico Kolter, and Zachary C. Lipton. RATT: Leveraging unlabeled data to guarantee generalization. *Proceedings of the 38th International Conference on Machine Learning*, PMLR 139:3598-3609, 2021.

---

> ### Author Response · Authors · 2024-12-02
> **A Gentle Reminder**
>
> Dear Reviewer R7ur,
>
> As the rebuttal phase nears its end, we want to gently remind you of our responses and would greatly appreciate your further feedback.
>
> In the above rebuttal, we have addressed all your concerns, comments and additional questions with the utmost care and detailed responses. Hope our efforts meet your expectations. If you feel your concerns have been adequately addressed and find our updates satisfactory, with the utmost respect, we invite you to consider a score adjustment.  If you have any remaining concerns, we would be happy to discuss them further, and we hope to make the most of the remaining day to provide further clarifications.
>
> We deeply appreciate the time and effort you have dedicated to reviewing our paper. Regardless of the final outcome, we want to express our heartfelt gratitude for your thoughtful feedback and contributions to improving our work. Thank you for your time and consideration!
>
> Best, \
> Authors

---

> > ### Comment · Reviewer_R7ur · 2024-12-03
> >
> > Thank you very much for your replies. I'm very sorry to answer so late -- the openreview emails are always classified as spam and thus I often miss them.
> >
> > Overall, I agree with most of your arguments and appreciate your clarifications. (Unfortunately, changes in the paper are not highlighted and the openreview diff does not work -- so, I cannot easily check what was changed in the paper.) I will go through the paper again in the next few days and have a close look again -- sorry that I wasn't able to do it before the discussion period ended.
> >
> > Nevertheless, I still believe that the novelty is rather limited and the empirical results do not fully convince me. Therefore, I will only increase my score to 5 for the moment.

---

> > > ### Author Response · Authors · 2024-12-04
> > > **Response to Reviewer R7ur**
> > >
> > > Thanks for your comments. We have outlined the novelty in details in the above rebuttal (**A1** in **Response to Reviewer R7ur**). Particularly, you can check **Table 1 (new)** in **Response to Reviewer G9Hm** for the newly updated core empirical results (large-scale unsupervised learning with distributed training).

---

### Official Review · Reviewer_G9Hm · 2024-11-02

**Soundness:** 3
**Presentation:** 3
**Contribution:** 2
**Rating:** 5
**Confidence:** 4

**Summary:**

The paper introduces a method called Evolved Sampling (ES) for efficient data selection in training machine learning models. The core contribution is a dynamic sampling framework that identifies informative data samples based on the evolution of loss values throughout the training process. By adjusting the selection of data at the batch level according to changes in loss values, ES significantly reduces the required training time while maintaining model accuracy.

**Strengths:**

1. Novelty - The paper introduces decoupled exponential moving averages, which leverage first-order loss differences for more stable and robust sampling, effectively combining ideas from loss and gradient-based sampling with robust optimization principles.

2. Quality - The paper provides theoretical proofs and experiments across models and datasets, demonstrating consistent gains in efficiency and robustness, especially under noisy labels.

3. Writing - The paper is clearly structured, with well-organized sections and visual aids that clarify ES’s advantages over traditional methods, though some theoretical sections may be dense for general readers.

4. Relevance -  ES offers practical relevance for reducing computational costs without accuracy loss, making it impactful for both research and industry applications in large-scale ML.

**Weaknesses:**

1. Significance - Much of the computation cost of foundation models occurs during pre-training, which is mostly self-supervised (auto-regressive, contrastive learning, auto-encoders). All the experiments in the paper are for labeled datasets, which represent fine-tuning use cases where the computation cost is not a major concern. Thus, the significance of the method is not clearly demonstrated.

2. Scalability - The paper claims that ES has only modest overheads, but lacks an in-depth analysis of computational and memory costs associated with the decoupled EMA calculations, especially in large-scale tasks or datasets.

3. Assumptions - Some assumptions in theoretical analysis may not hold in practice, e.g., smoothness of loss functions, especially for complex architectures and non-convex losses. A discussion of how the method performs when assumptions deviate from theory, or empirical analysis on non-smooth tasks, would help clarify the applicability.

4. Hyperparameter Sensitivity - Introducing 2 hyperparameters could be a major concern for the proposed method. The current analysis (Figure 5) is too limited, e.g., what's the impact of hyperparameters on efficiency? Besides, it does seem that hyperparameters introduce a large variance in performance. For fair comparisons, the cost of searching hyperparameters should also be considered in the overall task (e.g., on a smaller dataset to test hyperparameters and then apply to a large dataset.)

5. Lack of Baselines for Noise - In the experiments on label noise, ES performs well, but the comparison is limited mainly to non-specialized sampling methods.

nit - ES in this literature often refers to 'Evolution Strategy', so would be nice to have a different abbreviation for the proposed method.

**Questions:**

1. Could the authors provide more insight into the sensitivity of the hyperparameters $(\beta_1, \beta_2)$ across different datasets and architectures?

2. ES appears computationally feasible for single-machine training, but would its performance gains hold up in distributed training settings?

3. ES with Pruning (ESWP) combines batch and set-level selection, but it is not entirely clear how this combination impacts overall performance in practice.

4. How can ES be used for self-supervised training?

---

> ### Author Response · Authors · 2024-11-27
> **Response to Reviewer G9Hm**
>
> >**Q1**: Significance - Much of the computation cost of foundation models occurs during pre-training, which is mostly self-supervised (auto-regressive, contrastive learning, auto-encoders). All the experiments in the paper are for labeled datasets, which represent fine-tuning use cases where the computation cost is not a major concern. Thus, the significance of the method is not clearly demonstrated.
>
> **A1**: First, we clarify that the fine-tuning task studied in this work (Table 3) is for *full fine-tuning*, also possessing considerable computation costs when scaling to large models (ViT-Large) and datasets (ImageNet-1K). **In addition, we also add the corresponding pre-training experiments under the distributed learning setting as follows.**
>
> - Table 1: The re-construction loss and running time of (MAE-based) pre-training the ViT-Large model on the ImageNet-1K dataset for 300 epochs (with 4xA100).
>     | |Baseline|ES|ESWP
>     |----|----|----|----|
>     |Loss|$0.425$|$0.439$|$0.433$|
>     |Time (h)|$48.7$|$42.8$|$40.1$|
>
> >**Q2**: Scalability - The paper claims that ES has only modest overheads, but lacks an in-depth analysis of computational and memory costs associated with the decoupled EMA calculations, especially in large-scale tasks or datasets.
>
> **A2**: For computation:
> - We note that the additional computation only arises from forward passes, whose overhead is much less than that of backward passes. Therefore, reducing the backward propagation computation as in our method would be dominantly effective to accelerate the training. The acceleration effect has been reflected by overall reduced time, as is shown in extensive experiments in Section 4.1 in the original manuscript.
> - We claim that the additional computation introduced by forward passes is modest, since compared to the baseline (no data selection), one only needs to additionally compute the losses on *selected mini*-batches with reduced sizes compared to original (meta-)batches.
>
> For memory:
> - It is straightforward to deduce from Eq. (3.8) that the additional memory is $O(n)$ ($n$: sample size), since we need to store the score value of each data sample for only one single training step. The additional $O(n)$ memory costs are negligible since only $O(1)$ extra space is required for each data sample consisting of high-dimensional tensors.
> - **We numerically test the overall memory costs of ES(WP), which are shown to be reduced compared to the baseline (no data selection).**
>     - Table 2: The averaged memory usage under the default  configuration of batch sizes ($b=64$, $B=256$) when (full) fine-tuning the ViT-Large model on the ImageNet-1K dataset (with 1xA100 (80GB)).
>         | |Baseline|ES|ESWP
>         |----|----|----|----|
>         |Memory (GB)|$52.4$|$49.7$|$49.1$
>
> >**Q3**: Assumptions - Some assumptions in theoretical analysis may not hold in practice, e.g., smoothness of loss functions, especially for complex architectures and non-convex losses. A discussion of how the method performs when assumptions deviate from theory, or empirical analysis on non-smooth tasks, would help clarify the applicability.
>
> **A3**: We respond by two points:
> - We clarify that Proposition 1 is derived just aiming to theoretically *motivate* data selectiom methods based on loss re-weighting. With its mathematically proved convergence accelerations, and by viewing loss re-weighting as $0$-order EMA, we achieve analytical developments towards ES(WP) (Section 3.3). That is, one can naturally extend the data selection framework to EMAs with higher orders, i.e. standard EMA ($1$-order) and decoupled EMA ($2$-order) used in ES(WP).
> - In practice, we *do* provide empirical analysis in general settings. In fact, all experimental results under the "Loss" data selection method in Section 4 are desired results.

---

> ### Author Response · Authors · 2024-11-27
> **Response to Reviewer G9Hm (continue)**
>
> >**Q4**: Hyperparameter Sensitivity - Introducing 2 hyperparameters could be a major concern for the proposed method. The current analysis (Figure 5) is too limited, e.g., what's the impact of hyperparameters on efficiency? Besides, it does seem that hyperparameters introduce a large variance in performance. For fair comparisons, the cost of searching hyperparameters should also be considered in the overall task (e.g., on a smaller dataset to test hyperparameters and then apply to a large dataset.)
>
> **A4**: We plot Figure 5 to convey the main message that default configurations of hyperparameters $(\beta_1, \beta_2)$ are basically okay: The default betas are consistently validated to be roughly (locally) optimal in small-scale models and datasets, and their *superior* effectiveness and efficiency remain in other experiments under different settings (e.g. noisy supervision, Table 5 and Figure 3) and with larger scales (e.g. Table 3). **We also extend Figure 5 for denser betas around the default values to verify its (local) optimality.**
>
> - Table 3: Test accuracies (\%) for different betas in ES (ResNet-18, CIFAR-100).
>     |$\beta_2$ \ $\beta_1$|$0.15$|$0.2$|$0.25$
>     |----|----|----|----|
>     |$0.95$|$78.5$|$78.8$|$78.1$
>     |$0.9$|$78.4$|$\textbf{78.8}$|$78.6$
>     |$0.85$|$78.3$|$78.4$|$78.3$
>
> >**Q5**: Lack of Baselines for Noise - In the experiments on label noise, ES performs well, but the comparison is limited mainly to non-specialized sampling methods.
>
> **A5**: We point out that the scope of this work is within comparisons between general data selection methods. In fact, whether there are label noises or not and the portion of noises are often unknown in practical applications, hence data selection methods specialized for these noisy settings (if any) are reasonably not the first choices.
>
> >**Q6**: nit - ES in this literature often refers to 'Evolution Strategy', so would be nice to have a different abbreviation for the proposed method.
>
> **A6**: Thanks for the reminder. We plan to replace the method name with e.g. sampling by diff-loss re-weighting. Any further suggestions are welcomed.
>
> >**Q7**: Could the authors provide more insight into the sensitivity of the hyperparameters across different datasets and architectures?
>
>  **A7**: Please refer to **A4** for details.
>
> >**Q8**: ES appears computationally feasible for single-machine training, but would its performance gains hold up in distributed training settings?
>
> **A8**: Please refer to **A1** for details.
>
> >**Q9**: ES with Pruning (ESWP) combines batch and set-level selection, but it is not entirely clear how this combination impacts overall performance in practice.
>
> **A9**: By combining both the batch and set level data selection, we aim to achieve more aggressive data pruning. **We further perform ablation studies on the pruning ratio $\text{pr}$ to demonstrate its impacts on the overall performance.**
> - Table 4: The effect of pruning ratios in ESWP (ResNet-18, CIFAR-10).
>     |$\text{pr}$|$0.1$|$0.2$|$0.3$|$0.4$|
>     |----|----|----|----|----|
>     |accuracy (\%)|$95.3$|$95.3$|$95.2$|$94.9$
>
> >**Q10**: How can ES be used for self-supervised training?
>
> **A10**: Please refer to **A1** for details.

---

> > ### Comment · Reviewer_G9Hm · 2024-11-27
> >
> > I appreciate the authors' responses. My main concern remains that the proposed method's significance, either on performance or speed, is not convincing enough. I'm also unsure how to interpret the pre-training experiment results. The reconstruction loss does not really indicate downstream performance, and the training time is different, what is the purpose here?
> >
> > Overall, I believe the paper is not ready to be published at its current stage, so my score remains unchanged.

---

> ### Author Response · Authors · 2024-12-02
> **Response to Reviewer G9Hm**
>
> Thanks for your comments. To clarify, we rerun the above experiments in Table 1 with further fine-tuning steps on classification tasks. The results are as follows.
>
> - Table 1 (new): The test accuracies ($\\%$) and reduced time of (MAE-based) pre-training the ViT-Large model on the ImageNet-1K dataset for 300 epochs (with 4xA100), and fine-tuning for 50 epochs.
> | |Baseline|ES|ESWP
> |----|----|----|----|
> |Top-1 accuracy|$84.9$|$84.8$|$84.7$|
> |Top-5 accuracy|$97.2$|$97.2$|$97.1$|
> |Time $\downarrow$|$-$|$12.1\\%$|$17.7\\%$|
>
> It is observed that with fixed budgets of total training epochs (which is often required in practice to determine the learning rate schedule in advance, and "fixed total training epochs" is the typical default setting of many former related works, e.g. InfoBatch ([1]; ICLR 2024 Oral), which is possibly the most recent SOTA and also compares with *only "Baseline"*), ES(WP) can achieve comparable performance with the baseline, but lead to significant accelerations. In this experiment, the configuration of all hyper-parameters remains the same with other experiments in the manuscript, except that the mini-batch size is enlarged as $b=192$ (the meta-batch size is still $B = 256$). We hope that these newly updated results solve your concerns.
>
>
>
> **References**
>
> [1] Ziheng Qin, Kai Wang, Zangwei Zheng, Jianyang Gu, Xiangyu Peng, Zhaopan Xu, Daquan Zhou, Lei Shang, Baigui Sun, Xuansong Xie, and Yang You. InfoBatch: Lossless training speed up by unbiased dynamic data pruning. In *International Conference on Learning Representations*, 2024.

---

### Official Review · Reviewer_M8jH · 2024-11-02

**Soundness:** 3
**Presentation:** 3
**Contribution:** 2
**Rating:** 5
**Confidence:** 4

**Summary:**

The paper proposes "Evolved Sampling" (ES), a dynamic sampling method aimed at improving data efficiency during training. The method selects informative samples based on the loss values during training using a decoupled Exponential Moving Average (EMA) scheme. This reduces the number of samples needed for backpropagation, saving up to 40% in wall-clock time while maintaining model performance. The method was tested on a thorough evaluation across many different models (ResNet, ViT, ALBERT) and datasets (CIFAR, ImageNet, GLUE).

**Strengths:**

- ES shows a reduction in training time without loss in performance, which is promising for computationally expensive tasks.
- The use of loss evolution for sampling is an interesting approach that addresses the shortcomings of previous static and simple dynamic sampling methods.
- The results on datasets with noisy labels are interesting.
- Evaluation is sufficiently complete.

**Weaknesses:**

- Limited novelty: the paper largely builds on existing sampling concepts with incremental improvements.
- The description of the method can be simplified considerably.

- While the method helps reducing the number of backpropagation steps performed during training, it still requires feedforward running of all samples through the network, which is still computationally expensive. Indeed, while the results are positive, the measured gains are not particularly game-changing.

- Minor: I am not sure "evolved" is the right term here;  "evolved" and "ES" remind strongly of evolutionary optimization and "Evolution Strategies", which can introduce confusion.

- It would be interesting to read more about the increased robustness to label noise; I might have expected the proposed method to perform worse, since samples with wrong labels would yield higher losses (unless/until the network memorizes the whole training set).

**Questions:**

See Weaknesses.

---

> ### Author Response · Authors · 2024-11-27
> **Response to Reviewer M8jH**
>
> >**Q1**: Limited novelty: the paper largely builds on existing sampling concepts with incremental improvements.
>
> **A1**: The existing sampling concepts where this work is motivated by are mainly loss re-weighting and EMAs. However, the present work differentiates from them with at least three points as follows:
> - Loss re-weighting: Although the loss re-weighting is one of our motivations, its effectiveness was only numerically verified in applications and no theoretical characterizations were provided before. In this work, we first mathematically prove the convergence acceleration achieved by loss re-weighting (Proposition 1), which is certainly novel.
> - EMAs: Although the standard EMA has been adopted in data selection (e.g. UCB ([1])), its practical performance is not that satisfying (e.g. [1]). In this work, we propose a new EMA scheme (decoupled EMA), acting as a non-trivial extension in the following sense:
>     - In formulation, the decoupled EMA "space" contains former loss re-weighting based data selection methods. It is easy to see that decoupled EMA reduces to standard EMA by setting $\beta_1=\beta_2=\beta$, and further reduces to basic loss re-weighting and uniform sampling by setting $\beta=0$ and $\beta=1$, respectively.
>     - Intuitively, we illustrate the comparisons among dynamics of EMAs (Figure 1). It is shown that the decoupled EMA acts as an interpolation between the loss re-weighting and standard EMA: When losses oscillate, the decoupled EMA reacts moderately by not only capturing detailed dynamics of losses (which is ignored by standard EMA due to its over-smoothing effect), but also remaining certain robustness (while loss re-weighting is quite sensitive to loss variations). The decoupled EMA is also flexible to trade-off between these two regimes (details and smoothing) by tuning two betas (left $\to$ middle $\to$ right in Figure 1).
>     - In theory, we mathematically prove the novel Proposition 2 to demonstrate that the decoupled EMA is in fact a **first-order** modification of standard EMA, resulting in the flexibility to balance between losses and their **differences**. This potentially gives chances to better leverage the loss dynamics for data selection. Moreover, this first-order modification is *implicitly* introduced, meaning that only quantities involving losses are evaluated, without significant overheads as former first-order data selection methods based on gradients.
>     (This point is also recognized by other reviewers, say Reviewer G9Hm: "Novelty - The paper introduces decoupled exponential moving averages, which leverage *first-order* loss differences for more stable and robust sampling, effectively combining ideas from loss and gradient-based sampling with robust optimization principles.")
>     - Empirically, we observe non-trivial improvements in all experiments presented in Section 4, especially for datasets with noisy labels (*limited* investigations on this setting in former data selection references). Particularly, for the superiority of decoupled EMA over standard EMA, the theoretical insight derived in Proposition 2 is also numerically verified in Table 2, 3 and 5, where ES(WP) outperforms UCB, and abaltions in Table 6, where Loss + DE outperforms Loss + E for multiple architectures and datasets.
> - Additionally, the proposed method performs data selection in both the batch and set (epoch) level, which is also novel compared to former data selection literature (Table 1). This leads to more aggressive data pruning with better training accelerations and efficiency in appications.
>
> >**Q2**: The description of the method can be simplified considerably.
>
> **A2**: Currently, we aim to provide a complete description of our method. The method part is developed in a logical order as follows:
> - Preliminaries (Section 3.1): We present basic problem formulations.
> - Motivations (Section 3.2): We theoretically prove the convergence acceleration via loss re-weighting (Proposition 1), and mention several variants based on loss re-weighting in former references for completeness.
> - Analytical developments towards ES(WP) (Section 3.3): By viewing loss re-weighting as $0$-order EMA, we naturally extend the data selection framework to EMAs with higher orders, i.e. standard EMA ($1$-order) and decoupled EMA ($2$-order). Their intuitive comparisons are illustrated in Figure 1. Together with other general techniques (i.e. annealing and pruning), we finally achieve Algorithm 1 (illustrated in Figure 2).
>
> We would be appreciate if you can provide more *specific* suggestions to the description part, and we are certainly open to adjust the corresponding contents accordingly.

---

> ### Author Response · Authors · 2024-11-27
> **Response to Reviewer M8jH (continue)**
>
> >**Q3**: While the method helps reducing the number of backpropagation steps performed during training, it still requires feedforward running of all samples through the network, which is still computationally expensive. Indeed, while the results are positive, the measured gains are not particularly game-changing.
>
> **A3**: For the forward computation:
> - We point out that the additional computation introduced by forward passes is modest, since compared to the baseline (no data selection), one only needs to additionally compute the losses on *selected mini*-batches with reduced sizes compared to original (meta-)batches.
> - We note that the overhead of forward passes is much less than that of backward passes. Therefore, reducing the backward propagation computation as in our method would be dominantly effective to accelerate the training. The acceleration effect has been reflected by overall reduced time, as is shown in extensive experiments in Section 4.1 in the original manuscript.
>
> For the measured gains:
> - We have demonstrated the superiority of our method though extensive experiments in Seciton 4, by showing ES(WP)'s significant training accelerations on both small-scale and large-scale clean datasets (Seciton 4.1) and considerable learning accuracies enhancements on noisy datasets (Seciton 4.2).
> - Particularly, we emphasize that "InfoBatch" ([2]) is a strong data selection baseline (ICLR 2024 Oral, possibly the most recent SOTA), but our method surpasses InfoBatch in both learning accuracies and training accelerations (see Table 2, 3, 4, 5 and Figure 3(a)).
>
> >**Q4**: Minor: I am not sure "evolved" is the right term here; "evolved" and "ES" remind strongly of evolutionary optimization and "Evolution Strategies", which can introduce confusion.
>
> **A4**: Thanks for the reminder. We plan to replace the method name with e.g. sampling by diff-loss re-weighting. Any further suggestions are welcomed.
>
> >**Q5**: It would be interesting to read more about the increased robustness to label noise; I might have expected the proposed method to perform worse, since samples with wrong labels would yield higher losses (unless/until the network memorizes the whole training set).
>
> **A5**: In Figure 3(b), we plot the noisy gradient ratio (ngr) within training, i.e. illustrating the relative magnitudes of gradients evaluated on corrupted samples over the whole mini-batch along the training dynamics. This ratio is mathematically defined as $\text{ngr}:=\parallel\sum _ {i \in \mathfrak{b} _ t: y _ i=\tilde{y} _ i}\nabla _ {\theta} \ell _ {i}(\theta(t))\parallel _ 2/\parallel\sum _ {i \in \mathfrak{b} _ t}\nabla _ {\theta} \ell _ {i}(\theta(t))\parallel _ 2$ with $\mathfrak{b} _ t$ as the selected mini-batch for any training time $t$ and $\tilde{y}$ as the corrupted label.
>
> - As is shown in Figure 3, one can observe *strong dynamical correlations* between learning accuracies ($\text{acc}$) in Figure 3(a) and noisy gradient ratios ($\text{ngr}$) in Figure 3(b). That is, $\text{acc}$ always benefits from bounded $\text{ngr}$ and degrades from increased $\text{ngr}$, and their variations are almost *simultaneous* along the training dynamics (except Ordered SGD ("Order") due to its sensitivity with respect to losses (i.e., Ordered SGD always selects samples with top losses)).
> - However, despite that ES(WP) seems to select more noisy samples due to its loss re-weighting scheme, this intuition is not *provably* correct, since the variations in both the numerator and denominator of $\text{ngr}$ are indefinite due to offsets among terms.
> - In fact, as is shown in Figure 3, ES(WP) (and also basic loss re-weighting ("Loss")) really underperform at initial training stages. However, the $\text{acc}$ of ES(WP) and loss re-weighting somehow begins to upgrade (with non-increased $\text{ngr}$) and finally outperform. Since this phase transition occurs in the middle of training, the underlying mechanism may involve detailed learning dynamics, and the quantitative dynamical analysis can be quite difficult and complex, which is beyond the scope of the current algorithmic paper and left as the future work.
>
> **References**
>
> [1] Ravi Raju, Kyle Daruwalla, and Mikko Lipasti. Accelerating deep learning with dynamic data pruning. *arXiv preprint arXiv:2111.12621*, 2021.
>
> [2] Ziheng Qin, Kai Wang, Zangwei Zheng, Jianyang Gu, Xiangyu Peng, Zhaopan Xu, Daquan Zhou, Lei Shang, Baigui Sun, Xuansong Xie, and Yang You. InfoBatch: Lossless training speed up by unbiased dynamic data pruning. In *International Conference on Learning Representations*, 2024.

---

> ### Author Response · Authors · 2024-12-02
> **A Gentle Reminder**
>
> Dear Reviewer M8jH,
>
> As the rebuttal phase nears its end, we want to gently remind you of our responses and would greatly appreciate your further feedback.
>
> In the above rebuttal, we have addressed all your concerns, comments and additional questions with the utmost care and detailed responses. Hope our efforts meet your expectations. If you feel your concerns have been adequately addressed and find our updates satisfactory, with the utmost respect, we invite you to consider a score adjustment.  If you have any remaining concerns, we would be happy to discuss them further, and we hope to make the most of the remaining day to provide further clarifications.
>
> We deeply appreciate the time and effort you have dedicated to reviewing our paper. Regardless of the final outcome, we want to express our heartfelt gratitude for your thoughtful feedback and contributions to improving our work. Thank you for your time and consideration!
>
> Best, \
> Authors

---

### Meta-Review · Area_Chair_eeF2 · 2024-12-19

**Metareview:**

This paper proposes a method (ESWP) for supervised training, where in general, data points with higher losses are deemed more important for backpropagating, rather than regular batched gradient descent over uniformly chosen data points.

The core idea is to keep track of an Exponentially Moving Average (EMA) of data point importances (determined by loss vlaues), and then using these to determine which datapoints to actually be used for gradient descent. Specifically, Algorithm 1 is a loop consisting of:
  * "Pruning" down the entire dataset by performing importance sampling
  * Recomputing importance weights according to uniform batches using EMA updates.
  * "Annealing": Selecting minibatches according to their importances, OR sometimes performing uniform sampling again

Experimental results are conducted over:
  * Standard ResNet + CIFAR-10/100 settings with comparisons to other data selection methods
  * NLP tasks using ALBERT as a base model

Additional ablation studies are conducted to understand:
  * If there is label noise, the EMA allows for choosing the "right" data points
  * The importance of combining all effects (annealing and EMA)
  * Choices of method hyperparameters (batch size / minibatch sizes, EMA update coefficients).

## Strengths:
  * Experiments are fairly comprehensive and the method is solidly investigated.
  * Method is reasonable and well-motivated.

## Weaknesses:
  * Being blunt, the paper can easily fall into the category of "increasing complexity, but only gaining incremental improvements". The method introduces many new hyperparameters which may have to be tuned, and overall the improvements don't appear to be significant.
  * The paper doesn't provide us with any new profound lessons or conclusions.

**Additional Comments On Reviewer Discussion:**

The reviewer scores were a (5,5,5,8), signaling that the majority of the reviewers are lukewarm, with the exception of Reviewer 4oRD who gave the 8.

The most common and core issues are:
  * "Lack of novelty" - i.e. the method is mainly a combination of EMA, Annealing, and Pruning which have been investigated in previous papers.
  * Speedup gains - i.e. the method doesn't actually produce significant gains - e.g. slight percentage increases in accuracy on CIFAR-10, and slight percentage improvements in wall-clock time. The fairness of these results were also debated by multiple reviewers.

Reviewer 4oRD still wishes to champion the paper, on the basis of:
  * The paper doesn't need to be validated over new datasets (e.g. language modelling) since it requires large amounts of compute.
    * We can agree to this, although given the CIFAR-10 results provided already, it's not convincing that we would obtain huge gains in LLM training.
  * The paper raises the general notion that optimizing data selection and weights are both important.
    * As mentioned by other reviewers, this paper isn't the first to make this conclusion however, and it isn't new.

Seeing as most of the reviewers don't agree for acceptance, the decision is to reject.

---

### Decision · Program_Chairs · 2025-01-22

Reject